# SDP-CROWN: Efficient Bound Propagation for Neural Network Verification with Tightness of Semidefinite Programming

**Hong-Ming Chiu** [1]  **Hao Chen** [1]  **Huan Zhang** [1]  **Richard Y. Zhang** [1]

## Abstract

Neural network verifiers based on linear bound propagation scale impressively to massive models but can be surprisingly loose when neuron coupling is crucial. Conversely, semidefinite programming (SDP) verifiers capture inter-neuron coupling naturally, but their cubic complexity restricts them to only small models. In this paper, we propose SDP-CROWN, a novel hybrid verification framework that combines the tightness of SDP relaxations with the scalability of bound-propagation verifiers. At the core of SDP-CROWN is a new linear bound—derived via SDP principles—that explicitly captures $\ell_2$-norm-based inter-neuron coupling while adding only one extra parameter per layer. This bound can be integrated seamlessly into any linear bound-propagation pipeline, preserving the inherent scalability of such methods yet significantly improving tightness. In theory, we prove that our inter-neuron bound can be up to a factor of $\sqrt{n}$ tighter than traditional per-neuron bounds. In practice, when incorporated into the state-of-the-art $\alpha$-CROWN verifier, we observe markedly improved verification performance on large models with up to 65 thousand neurons and 2.47 million parameters, achieving tightness that approaches that of costly SDP-based methods.

## 1. Introduction

Neural network verification is critical for ensuring that models deployed in safety-critical applications adhere to robustness and safety requirements. Among various ver-

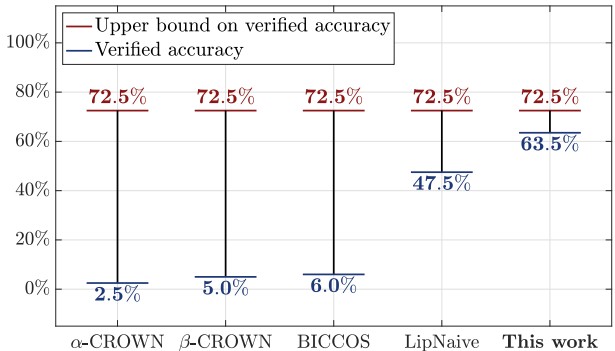

Figure 1: **Verification of the ConvLarge network on the CIFAR-10 dataset (six convolutional layers + three fully connected layers, $\approx$2.47M parameters and 65k neurons) under $\ell_2$ adversaries.** State-of-the-art bound propagation algorithms $\alpha$-CROWN, $\beta$-CROWN and BICCOS yield surprisingly loose relaxations under $\ell_2$ adversaries (verified accuracy 2.5%, 5.0% and 6.0%, respectively) at high cost (up to 289s per example). A naive Lipschitz baseline, which multiples per-layer $\ell_2$ Lipschitz constants directly ("LipNaive"), outperforms them (47.5% verified accuracy) with negligible runtime. In contrast, our proposed method achieves a striking 63.5% verified accuracy while keeping runtime moderate (73s). See Section 6.1 for experimental details; note that the model is too large to be verified with traditional SDP methods.

ification methods, linear bound propagation based approaches (Zhang et al., 2018; Wang et al., 2018; Wong & Kolter, 2018; Dvijotham et al., 2018; Singh et al., 2018; 2019; Xu et al., 2020) have emerged as the dominant approach due to their effectiveness and scalability. The core idea is to construct linear functions that provide pointwise upper and lower bounds for each nonlinear activation function and recursively propagate these bounds through the network. This method has proven particularly effective in certifying robustness against $\ell_\infty$-norm perturbations, where individual input features are perturbed within fixed limits. Notably, many highly ranked verifiers in VNN-COMP verification competition rely on bound propagation due to its success in scaling to large networks (Brix et al., 2023; 2024).

[1]Department of Electrical and Computer Engineering, University of Illinois at Urbana-Champaign. Correspondence to: Hong-Ming Chiu <hmchiu2@illinois.edu>, Hao Chen <haoc8@illinois.edu>, Huan Zhang <huan@huan-zhang.com>, Richard Y. Zhang <ryz@illinois.edu>.

*Proceedings of the $42^{nd}$ International Conference on Machine Learning*, Vancouver, Canada. PMLR 267, 2025. Copyright 2025 by the author(s).

Despite this success, bound propagation performs surprisingly poorly under $\ell_2$-norm perturbations, as shown in Figure 1. Unlike $\ell_\infty$-norm perturbations, which treat each neuron independently, $\ell_2$-norm perturbations impose inter-neuron coupling. This coupling introduces dependencies between neurons that bound propagation, designed to handle features individually, cannot effectively capture. As a result, it often produces loose and overly conservative output bounds. The $\ell_2$-norm setting is critical not only as a benchmark for evaluating neuron coupling (Szegedy et al., 2014) but also for verifying real-world adversarial examples, such as semantic perturbations, which are commonly modeled using $\ell_2$-norm perturbations applied through generative layers (Wong & Kolter, 2021; Barrett et al., 2022).

To overcome this limitation, semidefinite programming (SDP) methods (Raghunathan et al., 2018; Dathathri et al., 2020; Fazlyab et al., 2020; Anderson et al., 2021; Newton & Papachristodoulou, 2021; Chiu & Zhang, 2023) have been developed to explicitly model inter-neuron dependencies. These methods optimize over a dense $n \times n$ coupling matrix, yielding significantly tighter bounds compared to bound propagation. However, their cubic $O(n^3)$ time complexity restricts their application to relatively small models, and renders them impractical for realistic neural networks.

To bridge the gap between scalability and tightness, we introduce SDP-CROWN, a hybrid verification framework that combines the tightness of SDP with the efficiency of bound propagation. At the core of our framework is a linear bound derived through SDP principles that efficiently incorporates $\ell_2$-norm-based inter-neuron dependencies. A key feature of our bound is that it introduces only a single new parameter per layer, in contrast to traditional SDP methods, which require $n^2$ parameters per layer of $n$ neurons. As a result, our bound preserves the scalability of existing bound propagation methods. It can be seamlessly integrated into existing linear bound propagation verifiers such as CROWN and $\alpha$-CROWN, hence significantly tightening it for $\ell_2$ perturbations.

In theory, we prove that our proposed inter-neuron bound can be up to $\sqrt{n}$ times tighter than the per-neuron bounds commonly used in scalable verifiers. In practice, when incorporated into the $\alpha$-CROWN verifier, we find that SDP-CROWN significantly improves verification performance, achieving bounds close to those of expensive SDP-based methods while scaling to models containing over 65 thousand neurons and 2.47 million parameters. Our extensive experiments demonstrate that SDP-CROWN consistently enhances $\ell_2$ robustness certification rates across various architectures without compromising computational efficiency, making it well-suited for large-scale models where traditional SDP methods fail to scale.

## 1.1. Related work

To the best of our knowledge, this work is the first to apply SDP relaxations to efficiently tighten the linear bound propagation under $\ell_2$-norm perturbations.

SDP relaxation is the preferred approach for neural network verification against $\ell_2$-norm perturbations. Due to its ability to capture second-order information, SDP-based methods provide tight verification under $\ell_2$ adversaries (Chiu & Zhang, 2023). Several extensions have been proposed to further tighten the SDP relaxation by introducing linear cuts (Batten et al., 2021) and nonconvex cuts (Ma & Sojoudi, 2020), and to accommodate general activation functions (Fazlyab et al., 2020). However, SDP relaxation does not scale to medium-to-large scale models. Even with state-of-the-art SDP solvers and hardware acceleration (Dathathri et al., 2020; Chiu & Zhang, 2023), they remain computationally prohibitive for models containing more than 10 thousand neurons.

In addition to SDP relaxation, bound propagation methods (Zhang et al., 2018; Singh et al., 2018; 2019; Wang et al., 2018; Dvijotham et al., 2018; Hashemi et al., 2021; Xu et al., 2020; 2021) for certifying $\ell_2$ adversaries can be tightened using a branch-and-bound procedure (Wang et al., 2021; De Palma et al., 2021; Ferrari et al., 2022; Shi et al., 2025) by splitting unstable ReLU neurons into two subdomains, or by introducing nonlinear cutting planes (Zhang et al., 2022; Zhou et al., 2024) to capture the shape of $\ell_2$ adversaries. These methods have proven effective when the $\ell_2$-norm perturbation is small, as there are relatively fewer unstable neurons. However, they can become ineffective for larger perturbations, and can completely fail, as demonstrated in Figure 1.

Alternatively, $\ell_2$ adversaries can be certified by lower bounding the robustness margin (2) using the network's global Lipschitz constant. To estimate this constant, (Fazlyab et al., 2019) solve the Lipschitz constant estimation problem using SDP relaxations; (Huang et al., 2021; Leino et al., 2021; Hu et al., 2023) incorporate a Lipschitz upper bound during training; and (Li et al., 2019; Trockman & Kolter, 2021; Singla & Feizi, 2022; Meunier et al., 2022; Xu et al., 2022; Araujo et al., 2023) design neural network architectures that are provably 1-Lipschitz. While these methods perform well on networks with a small global Lipschitz constant, verifying robustness based solely on the Lipschitz constant can still be overly conservative, as illustrated in Figure 1.

## 1.2. Notations

We use the subscript $x_i$ to denote indexing. We use $\mathbf{1}$ to denote a column of ones. We use $\odot$ to denote elementwise multiplication. We use $e_i$ to denote $i$-th standard basis vec-

tor. We use $X \succeq 0$ to denote $X$ being positive semidefinite. We use $|\cdot|$ to denote the *elementwise* absolute value, and $\|\cdot\|_p$ to denote the vector $\ell_p$ norm.

## 2. Preliminaries

### 2.1. Problem description

Consider the task of classifying a data point $x \in \mathbb{R}^n$ as belonging to the $i$-th of $q$ classes using a $N$-layer feedforward neural network $f : \mathbb{R}^n \to \mathbb{R}^q$. The network aims to generate a prediction vector that takes on its maximum value at the $i$-th element, i.e., $e_i^T f(x) > e_j^T f(x)$ for all incorrect labels $j \neq i$. We define the neural network $f(x) = z^{(N)}$ recursively as

$$x^{(k)} = \mathrm{ReLU}(z^{(k)}), \;\; z^{(k)} = W^{(k)} x^{(k-1)}, \;\; x^{(0)} = x \quad (1)$$

for $k \in \{1, 2, \ldots, N\}$ where $\mathrm{ReLU}(x) = \max\{x, 0\}$ and $W^{(k)}$ denote weight matrices. Without loss of generality, we ignore biases.

Given an input $\hat{x}$ of truth class $i$, the problem of verifying the neural network $f$ to have no adversarial example $x \approx \hat{x}$ mislabeled as the incorrect class $j \neq i$ can be posed as:

$$d_j = \min_{x \in \mathcal{X}} c^T f(x) \quad \text{s.t.} \quad (1), \quad (2)$$

where $c = e_i - e_j$ and $\mathcal{X}$ is a convex input set that models the adversarial perturbations. Popular choices include the elementwise bound

$$\mathcal{B}_\infty(\hat{x}, \hat{\rho}) = \{x \mid |x_i - \hat{x}_i| \leq \hat{\rho}_i \text{ for all } i\}$$

and the $\ell_2$ norm ball

$$\mathcal{B}_2(\hat{x}, \rho) = \{x \mid \|x - \hat{x}\|_2 \leq \rho\},$$

where $\hat{x} \in \mathbb{R}^n$ is a center point, and $\hat{\rho} \in \mathbb{R}^n$ and $\rho \in \mathbb{R}$ are the radii. The resulting vector $d \in \mathbb{R}^q$ is a *robustness margin* against misclassification. If $d \geq 0$ over all of its elements, then there exists no adversarial example $x$ within a distance of $\rho$ that can be misclassified.

### 2.2. Semidefinite programming (SDP) relaxation

Semidefinite relaxation is a convex relaxation method to compute lower bounds on (2). In this work, we focus our attention on the SDP relaxation used in Brown et al. (2022) that utilizes the positive/negative splitting of the preactivations $u_i = x_i$, $v_i = x_i - z_i$ to rewrite the equality constraints $x_i = \mathrm{ReLU}(z_i)$ as

$$x_i = u_i, \quad u_i v_i = 0, \quad u_i \geq 0, \quad v_i \geq 0.$$

Adding $[1 \; u_i \; v_i]^T [1 \; u_i \; v_i] \succeq 0$ and relaxing $U_i = u_i^2$ and $V_i = v_i^2$ yields the SDP relaxation of the ReLU activation

$$x_i = u_i, \;\; u_i \geq 0, \;\; v_i \geq 0, \;\; \begin{bmatrix} 1 & u_i & v_i \\ u_i & U_i & 0 \\ v_i & 0 & V_i \end{bmatrix} \succeq 0. \quad (3)$$

Similarly, the SDP relaxation of $\mathcal{B}_2(\hat{x}, \rho)$ is given by

$$\sum_{i=1}^n U_i - 2(u_i - v_i)\hat{x}_i + V_i + \hat{x}_i^2 \leq \rho^2,$$

$$u_i \geq 0, \quad v_i \geq 0, \quad \begin{bmatrix} 1 & u_i & v_i \\ u_i & U_i & 0 \\ v_i & 0 & V_i \end{bmatrix} \succeq 0. \quad (4)$$

The SDP relaxation of (2) can be derived via (3) and (4). While SDP relaxations are typically tighter than most other convex relaxation methods, existing approaches solve the SDP relaxation via interior point method (Brown et al., 2022) or low-rank factorization method (Chiu & Zhang, 2023). Those methods incur approximately cubic time complexity and are not scalable to medium-scale models.

## 3. Looseness of bound propagation for $\ell_2$-norm perturbations

Linear bound propagation is one of the state-of-the-art approaches for finding upper and lower bounds on (2). In this section, we explain why the approach can be unusually loose when faced with an $\ell_2$ perturbation set like $\mathcal{X} = \mathcal{B}_2(\hat{x}, \rho)$, which is the classic example when interneuron coupling strongly manifests. For simplicity, we focus on finding a lower bound for (2).

At a high level, all bound propagation methods solve (2) by defining a set of linear relaxations $x \mapsto g^T x + h$ that *pointwise* lower bound the original function $c^T f(x)$ across the input set $\mathcal{X}$, as in

$$\mathscr{L}(\mathcal{X}) = \{(g, h) \mid c^T f(x) \geq g^T x + h \text{ for all } x \in \mathcal{X}\}.$$

The linear relaxation corresponding to each $(g, h) \in \mathscr{L}(\mathcal{X})$ can be minimized to yield a valid lower bound on the original problem (2). This bound can be further tightened by optimizing over the linear relaxations themselves:

$$\min_{x \in \mathcal{X}} c^T f(x) \geq \max_{(g,h) \in \mathscr{L}(\mathcal{X})} \min_{x \in \mathcal{X}} g^T x + h.$$

In fact, one can show by a duality argument that the bound above is in fact exactly tight, i.e. the inequality holds with equality. Unfortunately, the set of linear relaxations $\mathscr{L}(\mathcal{X})$ is also intractable to work with.

Instead, all bound propagation methods work by constructing parameterized families of linear relaxations $x \mapsto g(\alpha)^T x + h(\alpha)$ for $0 \leq \alpha \leq 1$ that provably satisfy $(g(\alpha), h(\alpha)) \in \mathscr{L}(\mathcal{X})$. The tightest bound on (2) that could be obtained from the family of relaxations then reads

$$\min_{x \in \mathcal{X}} c^T f(x) \geq \max_{0 \leq \alpha \leq 1} \min_{x \in \mathcal{X}} g(\alpha)^T x + h(\alpha). \quad (5)$$

Note that the inner minimization is a convex program that can be efficiently evaluated for many common choices of

$\mathcal{X}$, such as the elementwise bound or any $\ell_p$ norm ball. In practice, the parameter $\alpha$ can be maximized via projected gradient ascent or selected heuristically as in Zhang et al. (2018).

The tightness of the heuristic bound in (5) is critically driven by the quality of the parameterized relaxations $x \mapsto g(\alpha)^T x + h(\alpha)$. The core insight of bound propagation methods is that a high-quality choice of $g(\alpha), h(\alpha)$ satisfying the following

$$c^T f(x) \geq g(\alpha)^T x + h(\alpha) \text{ for all } x \in \mathcal{B}_\infty(\hat{x}, \hat{\rho})$$

can be constructed using the triangle relaxation of the ReLU activation, alongside a forward-backward pass through the neural network; we refer the reader to the Appendix B for precise details. When the input set is indeed an $\ell_\infty$-norm box $\mathcal{X} = \mathcal{B}_\infty(\hat{x}, \hat{\rho})$, Salman et al. (2019) showed that this choice of $(g(\alpha), h(\alpha)) \in \mathcal{L}(\mathcal{B}_\infty(\hat{x}, \hat{\rho}))$ is essentially *optimal* per-neuron. This optimality provides a long-sought explanation for the tightness of bound propagation under an $\ell_\infty$ adversary.

However, when the input set $\mathcal{X}$ is not an $\ell_\infty$-norm box, bound propagation requires relaxing the input set $\mathcal{X} \subseteq \mathcal{B}_\infty(\hat{x}, \hat{\rho})$ for the purposes of constructing $g(\alpha), h(\alpha)$. The resulting bound on (2) is valid by the following sequence of inequalities

$$\min_{x \in \mathcal{X}} c^T f(x) \geq \max_{(g,h) \in \mathcal{L}(\mathcal{X})} \min_{x \in \mathcal{X}} g^T x + h$$
$$\geq \max_{(g,h) \in \mathcal{L}(\mathcal{B}_\infty(\hat{x}, \hat{\rho}))} \min_{x \in \mathcal{X}} g^T x + h \quad (6)$$
$$\geq \max_{0 \leq \alpha \leq 1} \min_{x \in \mathcal{X}} g(\alpha)^T x + h(\alpha).$$

The problem is that a loose relaxation $\mathcal{B}_\infty(\hat{x}, \hat{\rho}) \supseteq \mathcal{X}$ causes a comparably loose relaxation $\mathcal{L}(\mathcal{B}_\infty(\hat{x}, \hat{\rho})) \subseteq \mathcal{L}(\mathcal{X})$ in (6), hence introducing substantial conservatism to the overall bound.

The above explains the core mechanism for why bound propagation tends to be loose for an $\ell_2$ adversary. The problem is that the tightest $\ell_\infty$-norm box to fully contain a given $\ell_2$-norm ball satisfies the following

$$\mathcal{X} = \mathcal{B}_2(\hat{x}, \rho) \subseteq \mathcal{B}_\infty(\hat{x}, \mathbf{1}\rho).$$

However, there are attacks in the box $x \in \{\pm\rho\}^n \subseteq \mathcal{B}_\infty(\hat{x}, \mathbf{1}\rho)$ with radii $\|x - \hat{x}\|_2 = \sqrt{n}\rho$ that are a factor of $\sqrt{n}$ larger than the radius $\rho$ of the original ball. Accordingly, relaxing the $\ell_2$-norm ball into $\ell_\infty$-norm box can effectively increase the attack radius by a factor of $\sqrt{n}$. Hence, the resulting bounds on (2) can also be a factor of $\sqrt{n}$ more conservative.

## 4. Proposed method

Our core contribution in this paper is a high-quality family of linear relaxations $x \mapsto g^T x + h(g, \lambda)$ for $g \in \mathbb{R}^n$ and $\lambda \geq 0$ that provably satisfy the following

$$c^T f(x) \geq g^T x + h(g, \lambda) \text{ for all } x \in \mathcal{B}_2(\hat{x}, \rho).$$

Notice that our relaxation is constructed directly from the $\ell_2$-norm ball, i.e. $(g, h(g, \lambda)) \in \mathcal{L}(\mathcal{X})$, which addresses the looseness of (6) as we did not relax the $\ell_2$-norm ball into $\ell_\infty$-norm box. Due to space constraints, we explain our construction only for the special case of $f(x) \equiv \text{ReLU}(x)$, while deferring the general case to the Appendix B.

One particle aspect of our construction is to take a linear relaxation from bound propagation $c^T f(x) \geq g(\alpha)^T x + h(\alpha)$ for the box $\mathcal{B}_\infty(\hat{x}, \rho\mathbf{1}) \supseteq \mathcal{B}_2(\hat{x}, \rho)$, and then tightening the offset $h(g(\alpha), \lambda) \geq h(\alpha)$ while ensuring that it remains valid for the ball $\mathcal{B}_2(\hat{x}, \rho)$. In analogy with Salman et al. (2019), we prove in Section 5 that this choice of $h(g(\alpha), \lambda)$ is essentially optimal when $\hat{x} = 0$, and can therefore yield at most a factor of $\sqrt{n}$ reduction in conservatism for $\mathcal{X} = \mathcal{B}_2(\hat{x}, \rho)$. At the same time, our new method adds just one parameter $\lambda \geq 0$ per layer, so it can be seamlessly integrated into any bound propagation verifier with negligible overhead. Integrating this technique into the $\alpha$-CROWN verifier, we provide extensive computational verification in Section 6 showing that our method significantly improves verification performance. The main theorem of our work is summarized below.

**Theorem 4.1.** *Given $c, \hat{x} \in \mathbb{R}^n$ and $\rho \geq 0$. The following holds*

$$c^T \text{ReLU}(x) \geq g^T x + h(g, \lambda) \text{ for all } x \in \mathcal{B}_2(\hat{x}, \rho)$$

*for any $\lambda \geq 0$ and $g \in \mathbb{R}^n$ where*

$$h(g, \lambda) = -\frac{1}{2} \cdot \left( \lambda(\rho^2 - \|\hat{x}\|_2^2) + \frac{1}{\lambda} \|\phi(g, \lambda)\|_2^2 \right)$$

*and*

$$\phi_i(g, \lambda) = \min\{c_i - g_i - \lambda\hat{x}_i, g_i + \lambda\hat{x}_i, 0\}.$$

Let us explain how Theorem 4.1 can be used to lower bound the attack problem (2) in the special case of $f(x) \equiv \text{ReLU}(x)$ and $\mathcal{X} = \mathcal{B}_2(\hat{x}, \rho)$. First, we use the standard bound propagation procedure to compute linear relaxations $x \mapsto g(\alpha)^T x + h(\alpha)$ that provably satisfy $(g(\alpha), h(\alpha)) \in \mathcal{L}(\mathcal{B}_\infty(\hat{x}, \rho\mathbf{1}))$ for $0 \leq \alpha \leq 1$. Then, we replace $h(\alpha)$ with the new choice $h(g(\alpha), \lambda)$ specified in Theorem 4.1 to ensure that $(g(\alpha), h(g(\alpha), \lambda)) \in \mathcal{L}(\mathcal{B}_2(\hat{x}, \rho))$ for $\lambda \geq 0$. Both $\alpha$ and $\lambda$ can then be optimized to provide a tighter relaxation. We note that the attack problem can also be lower bounded by directly optimizing over $g$ and $\lambda \geq 0$ (by treating $\alpha$ as unconstrained variables), and Theorem 4.1 can be

extended to handle more general input set $\mathcal{X}$ such as an ellipsoid. We provide more details for these extensions in the Appendix C.

In the remainder of this section, we provide a proof of Theorem 4.1.

## 4.1. Proof of Theorem 4.1

Given any $c, g \in \mathbb{R}^n$ the process of finding the tightest possible $h$ such that $c^T \operatorname{ReLU}(x) \geq g^T x + h$ holds within within $\mathcal{B}_2(\hat{x}, \rho)$ admits the following generic problem

$$\min_{x \in \mathbb{R}^n} \ c^T \operatorname{ReLU}(x) - g^T x \quad \text{s.t.} \quad \|x - \hat{x}\|_2^2 \leq \rho^2. \quad (7)$$

Applying the positive/negative splitting $x = u - v$ where $u, v \geq 0$ and $u \odot v = 0$ yields the following

$$\min_{u, v \in \mathbb{R}^n} \ c^T u - g^T (u - v)$$
$$\text{s.t.} \ \|u\|_2^2 - 2(u-v)^T \hat{x} + \|v\|_2^2 \leq \rho^2 - \|\hat{x}\|_2^2$$
$$u \geq 0, \quad v \geq 0, \quad u \odot v = 0.$$

Though (7) is nonconvex due to the product of the two variables $u$ and $v$, a tight lower bound can be efficiently approximated via SDP relaxation described in (3) and (4). The SDP relaxation of (7) reads:

$$\min_{u, v, U, V \in \mathbb{R}^n} \ c^T u - g^T (u - v)$$
$$\text{s.t.} \ (U + V)^T \mathbf{1} - 2(u-v)^T \hat{x} \leq \rho^2 - \|\hat{x}\|_2^2,$$
$$u \geq 0, \quad v \geq 0,$$
$$\begin{bmatrix} 1 & u_i & v_i \\ u_i & U_i & 0 \\ v_i & 0 & V_i \end{bmatrix} \succeq 0 \text{ for } i = 1, \ldots, n.$$

The SDP relaxation can be further simplified by applying Theorem 9.2 of (Vandenberghe & Andersen, 2015):

$$\min_{\tilde{u}, \tilde{v}, u, v, U, V \in \mathbb{R}^n} \ c^T u - g^T (u - v)$$
$$\text{s.t.} \ (U + V)^T \mathbf{1} - 2(u-v)^T \hat{x} \leq \rho^2 - \|\hat{x}\|_2^2,$$
$$u \geq 0, \quad v \geq 0, \quad \tilde{u} + \tilde{v} = 1,$$
$$\begin{bmatrix} \tilde{u}_i & u_i \\ u_i & U_i \end{bmatrix} \succeq 0, \quad \begin{bmatrix} \tilde{v}_i & v_i \\ v_i & V_i \end{bmatrix} \succeq 0$$

for $i = 1, \ldots, n$. Let $\lambda \in \mathbb{R}$ denote the dual variables of the first inequality constraints and $s, t, \mu \in \mathbb{R}^n$ denote the dual variable for $u \geq 0$, $v \geq 0$ and $\tilde{u} + \tilde{v} = 1$, respectively. The Lagrangian dual is given by:

$$\max_{\lambda, s, t, \mu} \ -\frac{1}{2} \cdot \left( \lambda(\rho^2 - \|\hat{x}\|_2^2) + \mu^T \mathbf{1} \right)$$
$$\text{s.t.} \ \begin{bmatrix} \mu_i & c_i - g_i - \lambda \hat{x}_i - s_i \\ c_i - g_i - \lambda \hat{x}_i - s_i & \lambda \end{bmatrix} \succeq 0,$$
$$\begin{bmatrix} \mu_i & g_i + \lambda \hat{x}_i - t_i \\ g_i + \lambda \hat{x}_i - t_i & \lambda \end{bmatrix} \succeq 0,$$
$$\lambda \geq 0, \quad s \geq 0, \quad t \geq 0, \quad \mu \geq 0,$$

for $i = 1, \ldots, n$. For a $2 \times 2$ matrix $X$, note that $X \succeq 0$ holds if and only if $\det(X) \geq 0$ and $\operatorname{diag}(X) \geq 0$. Applying this insight yields a second-order cone programming (SOCP) problem

$$\max_{\lambda, s, t, \mu} \ -\frac{1}{2} \cdot \left( \lambda(\rho^2 - \|\hat{x}\|_2^2) + \mu^T \mathbf{1} \right)$$
$$\text{s.t.} \ \lambda \mu_i \geq (c_i - g_i - s_i - \lambda \hat{x}_i)^2, \quad (8)$$
$$\lambda \mu_i \geq (g_i - t_i + \lambda \hat{x}_i)^2,$$
$$\lambda \geq 0, \quad s \geq 0, \quad t \geq 0, \quad \mu \geq 0,$$

for $i = 1, \ldots, n$. Due to space constraints, we defer the detailed derivation for the dual problem (8) to the Appendix D. We are now ready to prove Theorem 4.1.

*Proof.* Given any $c, g \in \mathbb{R}^n$. Let $\hat{\rho} = \rho^2 - \|\hat{x}\|_2^2$, $a_i = c_i - g_i - \lambda \hat{x}_i$ and $b_i = g_i + \lambda \hat{x}_i$. Fixing any $\lambda \geq 0$ and optimizing $\mu$ in (8) yields

$$\max_{\lambda, s, t \geq 0} \ -\frac{1}{2} \cdot \left( \lambda \hat{\rho} + \sum_{i=1}^n \frac{\max\left\{ (a_i - s_i)^2, (b_i - t_i)^2 \right\}}{\lambda} \right)$$
$$= \max_{\lambda \geq 0} \ -\frac{1}{2} \cdot \left( \lambda \hat{\rho} + \sum_{i=1}^n \frac{\min\left\{ a_i, b_i, 0 \right\}^2}{\lambda} \right)$$
$$= \max_{\lambda \geq 0} \ h(g, \lambda)$$

where the first equality follows from $\min_{s_i \geq 0} (a_i - s_i)^2 = \min\{a_i, 0\}^2$ and $\min_{t_i \geq 0} (b_i - t_i)^2 = \min\{b_i, 0\}^2$, and $\max\{\min\{a_i, 0\}^2, \min\{b_i, 0\}^2\} = \min\{a_i, b_i, 0\}^2$ for any $a_i, b_i \in \mathbb{R}$. Since $h(g, \lambda)$ is a lower bound on (7) for any $\lambda \geq 0$, we have $c^T \operatorname{ReLU}(x) \geq g^T x + h(g, \lambda)$ for all $x \in \mathcal{B}_2(\hat{x}, \rho)$ for any $g \in \mathbb{R}^n$, $\lambda \geq 0$. $\qquad \square$

## 5. Tightness analysis

In this section, we provide theoretical analysis on the tightness of our SDP relaxation in the special case where $f(x) \equiv \operatorname{ReLU}(x)$ and $\hat{x} = 0$. We show that our SDP relaxation (8) is exactly tight in this case and guarantees at most a factor of $\sqrt{n}$ improvement over bound propagation when computing linear relaxation within $\mathcal{B}_2(0, \rho)$. We begin by characterizing the linear relaxation from bound propagation $x \to g(\alpha)^T x + h(\alpha)$ for the box $x \in \mathcal{B}_\infty(0, \rho \mathbf{1}) \supseteq \mathcal{B}_2(0, \rho)$ below.

**Lemma 5.1.** *Given $c \in \mathbb{R}^n$ and $\rho > 0$, the bound*

$$g(\alpha) = \frac{1}{2} \min\{c, 0\} + \alpha \odot \max\{c, 0\},$$
$$h(\alpha) = -\rho \| \min\{g(\alpha), 0\} \|_1$$

*satisfies*

$$c^T \operatorname{ReLU}(x) \geq g(\alpha)^T x + h(\alpha) \ \text{for all} \ x \in \mathcal{B}_\infty(0, \rho \mathbf{1})$$

*for any $0 \leq \alpha \leq 1$.*

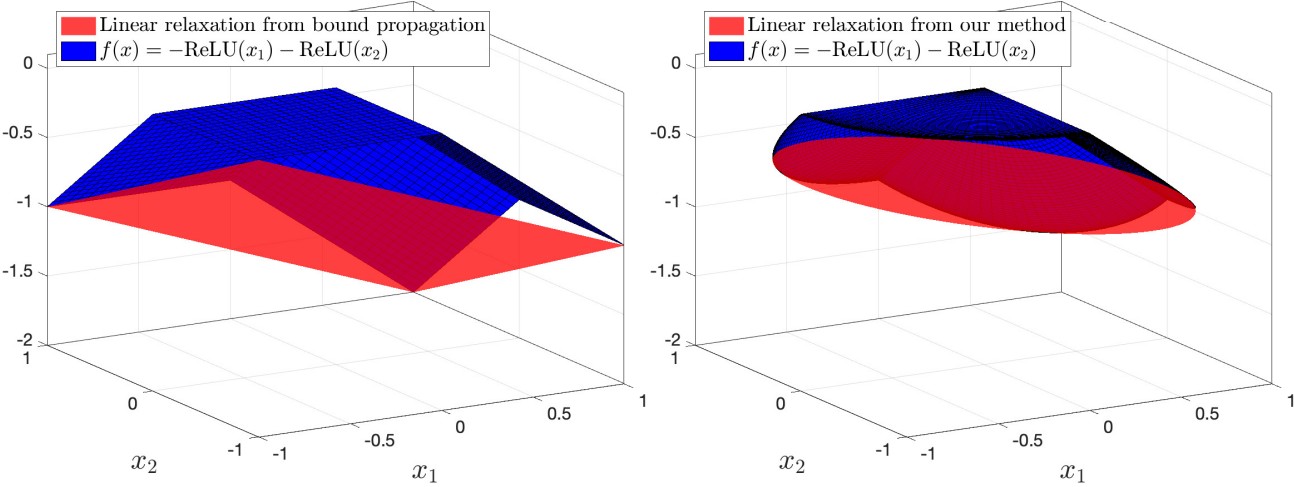

Figure 2: **Comparing linear relaxations within $\ell_2$-norm ball from our method and bound propagation.** Consider a task of finding a linear relaxation of $f(x) = -\mathrm{ReLU}(x_1) - \mathrm{ReLU}(x_2)$ on $\mathcal{B}_2(0,1)$. **(Left.)** Bound propagation finds the tightest possible linear relaxation on $\mathcal{B}_\infty(0,1)$, however, such relaxation is not the tightest on $\mathcal{B}_2(0,1)$. **(Right.)** Our method finds the tightest possible linear relaxation on $\mathcal{B}_2(0,1)$, which is tighter than bound propagation by a factor of $\sqrt{2}$.

*Proof.* Notice that $\alpha_i x_i \leq \mathrm{ReLU}(x_i) \leq \frac{1}{2}x_i + \frac{\rho}{2}$ for any $0 \leq \alpha_i \leq 1$. Therefore, we have $g(\alpha) = \frac{1}{2}\min\{c, 0\} + \alpha \odot \max\{c, 0\}$ and $h(\alpha) = \rho \sum_{i=1}^{n} \frac{1}{2}\min\{c_i, 0\} = -\rho \sum_{i=1}^{n} |\min\{g_i(\alpha), 0\}| = -\rho\|\min\{g(\alpha), 0\}\|_1$. $\square$

In the following Theorem, we show that our method yields $h(g(\alpha), \lambda) = -\rho\|\min\{g(\alpha), 0\}\|_2$ when $\lambda \geq 0$ is chosen optimally. On the other hand, the linear relaxation from bound propagation yields $h(\alpha) = -\rho\|\min\{g(\alpha), 0\}\|_1 \leq h(g(\alpha), \lambda)$ as in Lemma 5.1, which is looser than our method by at most a factor of $\sqrt{n}$.

**Theorem 5.2.** *Given $c \in \mathbb{R}^n$ and $\rho > 0$. The following holds*

$$c^T \mathrm{ReLU}(x) \geq g^T x - \rho\|\min\{c - g, g, 0\}\|_2$$

*for all $x \in \mathcal{B}_2(0, \rho)$ for any $g \in \mathbb{R}^n$.*

*Proof.* Setting $\hat{x} = 0$ in $h(g, \lambda)$ and optimizing over $\lambda \geq 0$ yields

$$\max_{\lambda \geq 0} -\frac{1}{2} \cdot \left( \lambda \rho^2 + \frac{1}{\lambda}\|\min\{c - g, g, 0\}\|_2^2 \right)$$
$$= -\rho\|\min\{c - g, g, 0\}\|_2$$

where the equality follows from $2\sqrt{ab} = \min_{x \geq 0} ax + b/x$ for any $a, b \geq 0$. $\square$

We obtain $h(g(\alpha), \lambda) = -\rho\|\min\{g(\alpha), 0\}\|_2$ by substituting $g = g(\alpha)$ in Theorem 5.2. Notice that $\min\{g(\alpha), 0\} = \min\{c - g(\alpha), g(\alpha), 0\}$ from Lemma 5.1. Theorem 5.2 guarantees at most a factor of $\sqrt{n}$ improvement over bound

propagation when $\hat{x} = 0$, which we provide a simple illustration in Figure 2.

Finally, we show that our SDP relaxation (8) is exactly tight in the following Theorem, i.e., both (8) and (7) attain the same optimal value. Therefore, the choice of $h(g(\alpha), \lambda)$ is optimal.

**Theorem 5.3.** *Given $c \in \mathbb{R}^n$ and $\rho > 0$. The following holds*

$$-\rho\|\min\{c - g, g, 0\}\|_2 = \min_{\|x\|_2 \leq \rho} c^T \mathrm{ReLU}(x) - g^T x$$

*for any $g \in \mathbb{R}^n$.*

*Proof.* Applying the positive/negative splitting $x = u - v$ where $u, v \geq 0$ and $u \odot v = 0$ yields

$$\min_{\substack{u,v \geq 0, \\ u \odot v = 0}} \sum_{i=1}^{n} (c_i - g_i)u_i + g_i v_i \quad \text{s.t.} \quad \sum_{i=1}^{n} u_i^2 + v_i^2 \leq \rho^2.$$

For each $i$, we substitute a variable $y_i$ according to the following three cases. Case 1: $c_i - g_i \leq \min\{g_i, 0\}$, we have $u_i^\star \geq 0$ and $v_i^\star = 0$; therefore we set $y_i = u_i$. Case 2: $g_i \leq \min\{c_i - g_i, 0\}$, we have $u_i^\star = 0$ and $v_i^\star \geq 0$; therefore we set $y_i = v_i$. Case 3: $0 \leq \min\{c_i - g_i, g_i\}$, we have $u_i^\star = v_i^\star = 0$; therefore we simply let $y_i \geq 0$. Substituting each $y_i$ yields

$$\min_{y \geq 0} \sum_{i=1}^{n} y_i \cdot \min\{c_i - g_i, g_i, 0\} \quad \text{s.t.} \quad \|y\|_2 \leq \rho,$$

which attains optimal value $-\rho\|\min\{c - g, g, 0\}\|_2$. $\square$

Table 1: **Verified accuracy under $\ell_2$-norm perturbations.** We report the verified accuracy (%) for 200 images. For each method, we also report the average verification time (in seconds or hours), except for LipNaive and LipSDP, where we report the total time for computing the Lipschitz constant. The upper bound on verified accuracy is estimated using projected gradient descent. A dash "-" indicates the model could not be evaluated due to excessive computational time.

| | Upper Bound | SDP-CROWN (Ours) | GCP-CROWN | BICCOS | $\beta$-CROWN | $\alpha$-CROWN | LipNaive | LipSDP (split=2) | LipSDP (no split) | LP-All | BM-Full |
|---|---|---|---|---|---|---|---|---|---|---|---|
| | | | | | **MNIST Model**[†] | | | | | | |
| MLP | 54% | 32.5% (2.5s) | 41% (173s) | 38% (198s) | 36% (302s) | 1.5% (1.2s) | 29% (0.02s) | 29.5% (19s) | 30.5% (62s) | 9% (75s) | **53%** (0.3h) |
| ConvSmall | 84.5% | **81.5%** (12s) | 19.5% (248s) | 17% (265s) | 16% (257s) | 0% (17s) | 77.5% (0.1s) | 78% (875s) | 78.5% (0.9h) | 10% (0.6h) | - |
| ConvLarge | 84% | **79.5%** (88s) | 0% (309s) | 0% (304s) | 0% (307s) | 0% (66s) | 77% (1s) | - | - | - | - |
| | | | | | **CIFAR-10 Model**[‡] | | | | | | |
| CNN-A | 55.5% | **49%** (12s) | 20% (210s) | 20% (224s) | 20% (201s) | 7.5% (3.8s) | 39% (0.2s) | 39% (1.7h) | - | - | - |
| CNN-B | 59.5% | **49.5%** (16s) | 3% (290s) | 3% (193s) | 3% (193s) | 0% (8.7s) | 33% (0.3s) | - | - | - | - |
| CNN-C | 47% | **42.5%** (10s) | 35.5% (96s) | 36% (101s) | 35.5% (63s) | 24.5% (5.8s) | 36.5% (0.2s) | 37% (0.5h) | 38.5% (1h) | 24.5% (0.3h) | - |
| ConvSmall | 52.5% | **43.5%** (9s) | 18% (225s) | 18% (220s) | 17.5% (146s) | 6% (4.4s) | 33% (0.2s) | 33.5% (1.2h) | - | - | - |
| ConvDeep | 50.5% | **46%** (25s) | 31% (133s) | 31.5% (133s) | 30.5% (133s) | 22.5% (9.2s) | 39.5% (0.3s) | 39.5% (1.9h) | - | - | - |
| ConvLarge | 72.5% | **63.5%** (73s) | 6% (286s) | 6% (282s) | 5% (289s) | 2.5% (47s) | 47.5% (1.2s) | - | - | - | - |

[†] The $\ell_2$-norm perturbation is set to be $\rho = 1.0$ for MLP, and $\rho = 0.3$ for both ConvSmall and ConvLarge.
[‡] The $\ell_2$-norm perturbation is set to be $\rho = 8/255$ for ConvLarge, and $\rho = 24/255$ for all the other models.

For the general case with $\hat{x} \neq 0$ and network $f(x)$ defined in (1), the improvement of our method cannot be analyzed analytically; instead, we present empirical validation in Section 6.3 to show that our method can be tighter than bound propagation in general settings.

# 6. Experiments

In this section, we compare the practical performance of our proposed method against several state-of-the-art neural network verifiers for certifying $\ell_2$ adversaries. The source code of our proposed method is available at https://github.com/Hong-Ming/SDP-CROWN.

**Methods.** SDP-CROWN denotes our proposed method. The implementation details for SDP-CROWN can be found in Appendix B. We compare SDP-CROWN against the following verifiers based on bound propagation: $\alpha$-CROWN (Xu et al., 2021), a bound propagation verifier with gradient-optimized bound propagation; $\beta$-CROWN (Wang et al., 2021), a verifier based on $\alpha$-CROWN that can additionally handle split constraints for ReLU neurons; GCP-CROWN (Zhang et al., 2022) and BICCOS (Zhou et al., 2024), verifiers based on $\beta$-CROWN that can additionally handle general cutting plane constraints. Since GCP-CROWN and BICCOS use mixed-integer programming (MIP) solvers to find cutting planes, we add the $\ell_2$-norm constraint into the MIP formulation of (2), so that all cutting planes generated from the MIP will consider the $\ell_2$-norm constraint rather than the enclosing $\ell_\infty$-norm constraint. We defer the detailed hyperparameter settings for bound propagation methods to the Appendix A.

We also compare SDP-CROWN against the following verifiers based on estimating an upper bound on the global Lipschitz constant: LipNaive, a verifier estimates the Lipschitz upper bound using the Lipschitz constant of each layer (as in Section 3 of Gouk et al. (2021)); and LipSDP (Fazlyab et al., 2019), a verifier estimates the Lipschitz upper bound based on solving SDP relaxations of the Lipschitz constant estimation problem. Specifically, LipNaive lower bounds $c^T f(x)$ within $\mathcal{B}_2(\hat{x}, \rho)$ through $c^T f(x) \geq c^T f(\hat{x}) - \rho \cdot \|c^T W^{(N)}\|_2 \cdot \|W^{(N-1)}\|_2 \cdots \|W^{(1)}\|_2$ where $\|W\|_2$ denotes the spectral $\ell_2$-norm of a matrix $W$.

Finally, we compare SDP-CROWN against the following verifiers based on directly solving convex relaxations of the verification problem (2): LP-All (Salman et al., 2019), a verifier solves an LP relaxation of (2) that uses the tightest possible preactivation bounds found by recursively solving LP problems for each preactivation; and BM-Full (Chiu & Zhang, 2023), a verifier solves an SDP relaxation of (2) that uses the same preactivation bounds in LP-All. For LP-All and BM-Full, we use the $\ell_2$-norm as the input constraint, as in $x \in \mathcal{B}_2(\hat{x}, \rho)$. The complexity of BM-Full and LP-All is cubic with respect to the number of preactivations, and therefore they are not scalable to most of the models used in our experiment.

**Setups.** All the experiments are run on a machine with a single Tesla V100-SXM2 GPU (32GB GPU memory) and dual Intel Xeon Gold 6138 CPUs.

**Models.** All the model architectures used in our experiment are taken from Wang et al. (2021) and Leino et al. (2021). To ensure non-vacuous $\ell_2$-norm robustness and make meaningful comparisons across different verification methods, we retrain all models to have a small global Lipschitz upper bound while keeping their model architecture unchanged. We defer the detailed model architecture and the training procedure to the Appendix A.

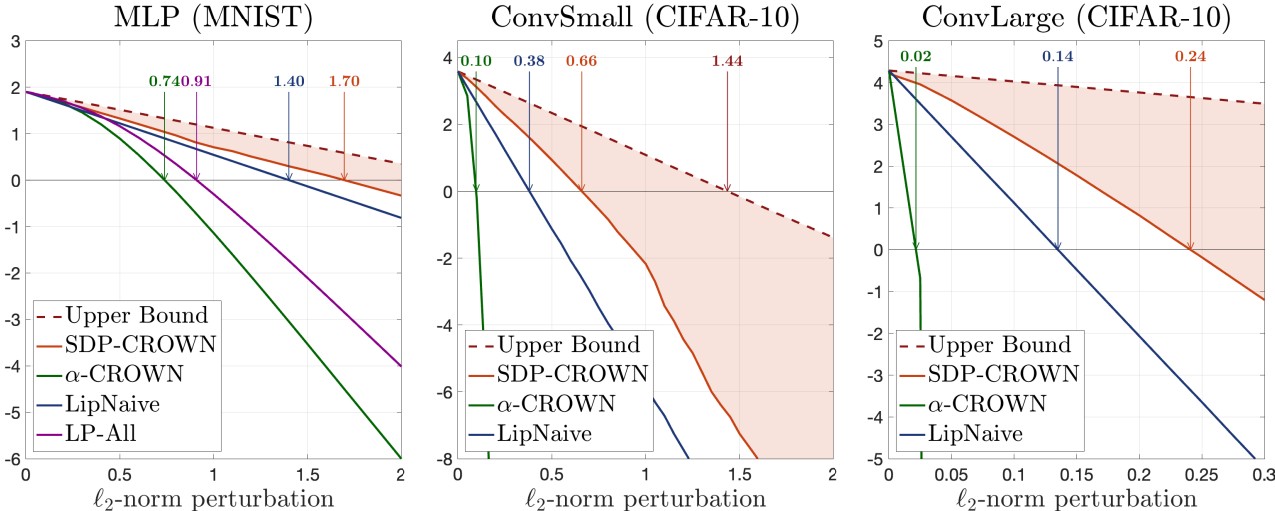

Figure 3: **Lower bounds on the robustness margin under $\ell_2$-norm perturbations**. We compare the lower bounds on (2) computed from SDP-CROWN, $\alpha$-CROWN, LipNaive and LP-All. The lower bounds are averages over 90 instances of (2). The upper bound on (2) is estimated projected gradient descent (PGD). The numbers in the figure indicate the $\ell_2$-norm perturbation level at which each lower bound crosses zero. Note that robustness verification is only meaningful in the interval where the PGD upper bound remains positive. **(Left.)** Small-scale model MLP (MNIST). **(Middle.)** Medium-scale model ConvSmall (CIFAR-10). **(Right.)** Large-scale model ConvLarge (CIFAR-10).

### 6.1. Robustness verification for neural networks

We compare the verified accuracy of SDP-CROWN against state-of-the-art verifiers on models trained on the MNIST and CIFAR-10 datasets. In each case, we fix the $\ell_2$-norm perturbation $\rho$ and compute verified accuracy using the first 200 images in the test set. Here, the verified accuracy denotes the percentage of inputs that are both correctly classified and robust. For comparison, we also compute the upper bound on verified accuracy using projected gradient descent (PGD) attacks (Madry et al., 2018).

Table 1 shows the verified accuracy and the average computation time for SDP-CROWN, GCP-CROWN, BICCOS, $\beta$-CROWN, $\alpha$-CROWN, LipNaive, LipSDP, LP-All and BM-Full. While BM-Full achieves the best verified accuracy in the first case, it unfortunately becomes computationally prohibitive in all remaining cases as its complexity scales cubically with respect to the number of preactivations. In all the remaining cases, SDP-CROWN achieves verified accuracy close to the PGD upper bound while the bound propagation method $\alpha$-CROWN exhibits limited certification performance. Other bound propagation methods $\beta$-CROWN, GCP-CROWN and BICCOS can greatly improve over $\alpha$-CROWN, but the gap is still large compared to SDP-CROWN, LipNaive and LipSDP.

Notably, SDP-CROWN is consistently tighter than Lip-Naive and LipSDP, suggesting that verifying robustness solely through the Lipschitz constant can be overly conser-

vative, even for networks trained to have a small Lipschitz constant. For LipSDP, networks are divided into subnetworks when evaluating the Lipschitz constant: split=1 indicates single-layer subnetworks, split=2 denotes two-layer subnetworks, and no split means the full network is evaluated directly. We ignore reporting split=1 for LipSDP as it only marginally improves over LipNaive.

Finally, LP-All is not scalable to large models and is also noticeably weaker than us on certified accuracy.

### 6.2. Tightness of lower bounds and verified accuracy

As the neural network verification problem (2) is NP-hard, all methods based on finding a lower bound on (2) via any sort of convex relaxations must become loose for sufficiently large $\ell_2$ perturbations. However, robustness verification is only necessary in the interval where the upper bound of (2) is positive, which can be efficiently estimated via PGD. Therefore, it is crucial for the lower bound to be tight within this region.

In this experiment, we examine the gap between the lower bound computed from SDP-CROWN and the upper bound computed from PGD across a wide range of $\ell_2$ perturbations. To ensure an accurate evaluation, we compute the average lower bounds over 90 instances of (2), which are generated via 9 incorrect classes for the first 10 correctly classified test images. We compare our average lower bound to $\alpha$-CROWN, LipNaive and LP-All, which

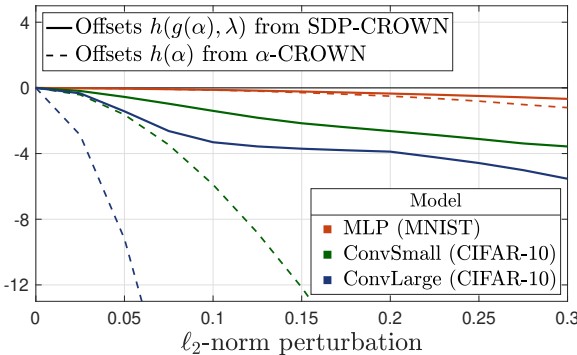

Figure 4: **Linear relaxation offsets under $\ell_2$-norm perturbations.** We compare the offset $h(\alpha)$ from $\alpha$-CROWN to the offset $h(g(\alpha), \lambda)$ from SDP-CROWN. The offsets are averages over 90 instances of (2). **(Red line.)** Small-scale model MLP (MNIST). **(Green line.)** Medium-scale model ConvSmall (CIFAR-10). **(Blue line.)** Large-scale model ConvLarge (CIFAR-10).

are verifiers also based on solving convex relaxations of (2). Figure 3 reports the average lower bounds and the average PGD upper bound with respect to three models of different scales: a small-scale model MLP (MNIST); a medium-scale model ConvSmall (CIFAR-10); and a large-scale model ConvLarge (CIFAR-10).

As shown in Figure 3, our proposed SDP-CROWN consistently outperforms $\alpha$-CROWN, LipNaive and LP-All, and significantly narrows the gap between the PGD upper bound. Notice that $\alpha$-CROWN produces extremely loose lower bounds in almost all cases, especially for large networks such as ConvLarge (CIFAR-10), where its lower bound drops rapidly. LP-All marginally improves over $\alpha$-CROWN. We note that LP-All is excluded in ConvSmall (CIFAR-10) and ConvLarge (CIFAR-10) due to its high computational cost. LipNaive provides tighter lower bounds than $\alpha$-CROWN and LP-All as all three models are trained to have small global Lipschitz upper bounds, but is consistently looser than SDP-CROWN.

### 6.3. Tightening bound propagation

In Section 5, we prove that when $f(x) \equiv \text{ReLU}(x)$ and the center of the $\ell_2$ norm perturbation is zero, the offset $h(g(\alpha), \lambda)$ computed from SDP-CROWN is guaranteed to be at least a factor of $\sqrt{n}$ tighter than the offset $h(\alpha)$ from bound propagation methods. However, the amount of improvement cannot be analyzed analytically under general settings. To empirically demonstrate how much improvement SDP-CROWN can achieve, in this experiment, we compute the average offset $h(g(\alpha), \lambda)$ from SDP-CROWN, and the average offset $h(\alpha)$ from $\alpha$-CROWN.

Specifically, the average offset of SDP-CROWN is taken over $h^{(k)}(g(\alpha^{(k)}), \lambda^{(k)})$ in (14) for $k = 1, \ldots, N$, as in $\frac{1}{N} \sum_{k=1}^{N} h^{(k)}(g(\alpha^{(k)}), \lambda^{(k)})$, and the average offset of $\alpha$-CROWN is taken over $h^{(k)}(\alpha^{(k)})$ in (12)for $k = 1, \ldots, N$, as in $\frac{1}{N} \sum_{k=1}^{N} h^{(k)}(\alpha^{(k)})$. To ensure an accurate evaluation, we compute the average offsets over 90 instances of (2), which are generated via 9 incorrect classes for the first 10 correctly classified test images. Figure 4 reports the average offsets with respect to three models of different scales: a small-scale model MLP (MNIST); a medium-scale model ConvSmall (CIFAR-10); and a large-scale model ConvLarge (CIFAR-10).

As shown in Figure 4, SDP-CROWN consistently yields a tighter offset compared to $\alpha$-CROWN under general settings, which demonstrates its effectiveness in tightening bound propagation and improves certification quality. Notably, $\alpha$-CROWN exhibits a significant drop in $h(\alpha)$ for larger networks such as ConvLarge (CIFAR-10) and ConvSmall (CIFAR-10) as the $\ell_2$ perturbation increases. In contrast, SDP-CROWN does not experience a rapid reduction in $h(g(\alpha), \lambda)$, maintaining significantly larger offsets across all models and perturbation sizes.

## 7. Conclusion

In this work, we present SDP-CROWN, a novel framework that significantly tightens bound propagation for neural network verification under $\ell_2$-norm perturbations. SDP-CROWN leverages semidefinite programming relaxations to improve the tightness of bound propagation while retaining the efficiency of bound propagation methods. Theoretically, we prove that SDP-CROWN can be up to $\sqrt{n}$ tighter than bound propagation for a one-neuron network under zero-centered $\ell_2$-norm perturbations. Practically, our extensive experiments demonstrate that SDP-CROWN consistently outperforms state-of-the-art verifiers across a range of models under $\ell_2$-norm perturbations, including models with over 2 million parameters and 65,000 neurons, where traditional LP and SDP methods are computationally infeasible, and bound propagation methods yield notably loose relaxations. Our results establish SDP-CROWN as both a theoretical and practical advancement in scalable neural network verification under $\ell_2$-norm perturbations.

## Acknowledgments

Financial support for this work was provided by NSF CAREER Award ECCS-2047462, IIS-2331967, and ONR Award N00014-24-1-2671. Huan Zhang is supported in part by the AI2050 program at Schmidt Sciences (AI2050 Early Career Fellowship).

## Impact Statement

The work presented in this paper aims to advance neural network verification in machine learning. There are many potential societal consequences of our work, none of which we feel must be specifically highlighted here.

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

## A. Experimental Setup

**Hyperparameter settings.** In our method SDP-CROWN, the variables $\alpha$ and $\lambda$ are solved by the Adam optimizer with 300 iterations. The learning rate is set to 0.5 and 0.05 for $\alpha$ and $\lambda$, respectively, and is decayed with a factor of 0.98 per iteration. For $\alpha$-CROWN, the variable $\alpha$ is solved by the Adam optimizer for 300 iterations, with the learning rate set to 0.5 and the decay factor set to 0.98. For $\beta$-CROWN, GCP-CROWN, and BICCOS, the timeout threshold for branch and bound is set to 300 seconds.

**Model architecture.** All the model architectures are taken from Wang et al. (2021) and Leino et al. (2021).

Table 2: Model architectures used in our experiments.

| Model name | Model structure | Parameters | Neurons | Accuracy |
|---|---|---|---|---|
| MLP (MNIST) | Linear(784, 100) - Linear(100, 100) - Linear(100, 10) | 89,610 | 994 | 79% |
| ConvSmall (MNIST) | Conv(1, 16, 4, 2, 1) - Conv(16, 32, 4, 2, 1) - Linear(1568, 100) - Linear(100, 10) | 166406 | 5598 | 87.5% |
| ConvLarge (MNIST) | Conv(1, 32, 3, 1, 1) - Conv(32, 32, 4, 2, 1) - Conv(32, 64, 3, 1, 1) - Conv(64, 64, 4, 2, 1) - Linear(3136, 512) - Linear(512, 512) - Linear(512, 10) | 1976162 | 48858 | 89.5% |
| CNN-A (CIFAR-10) | Conv(3, 16, 4, 2, 1) - Conv(16, 32, 4, 2, 1) - Linear(2048, 100) - Linear(100, 10) | 214918 | 9326 | 62% |
| CNN-B (CIFAR-10) | Conv(3, 32, 5, 2, 0) - Conv(32, 128, 4, 2, 1) - Linear(8192, 250) - Linear(250, 10) | 2118856 | 15876 | 66% |
| CNN-C (CIFAR-10) | Conv(3, 8, 4, 2, 0) - Conv(8, 16, 4, 2, 0) - Linear(576, 128) - Linear(128, 64) - Linear(64, 10) | 85218 | 5650 | 51% |
| ConvSmall (CIFAR-10) | Conv(3, 16, 4, 2, 0) - Conv(16, 32, 4, 2, 0) - Linear(1152, 100) - Linear(100, 10) | 125318 | 7934 | 60.5% |
| ConvDeep (CIFAR-10) | Conv(3, 8, 4, 2, 1) - Conv(8, 8, 3, 1, 1) - Conv(8, 8, 3, 1, 1) - Conv(8, 8, 4, 2, 1) - Linear(512, 100) - Linear(100, 10) | 54902 | 9838 | 53% |
| ConvLarge (CIFAR-10) | Conv(3, 32, 3, 1, 1) - Conv(32, 32, 4, 2, 1) - Conv(32, 64, 3, 1, 1) - Conv(64, 64, 4, 2, 1) - Linear(4096, 512) - Linear(512, 512) - Linear(512, 10) | 2466858 | 65546 | 74% |

*Note:* Conv(3, 16, 4, 2, 0) stands for a convolutional layer with 3 input channels, 16 output channels, a $4 \times 4$ kernel, stride 2 and padding 0. Linear(1568, 100) represents a fully connected layer with 1568 input features and 100 output features. There are no max pooling / average pooling used in these models and ReLU activations are applied between consecutive layers.

**Training procedure.** We retrained all the models used in our experiment to make them 1-lipschitz. The training strategy involves two phases: First, we train the model using the standard cross-entropy (CE) loss. Then, we retrain the model from scratch using a combination of KL-divergence and the spectral norm of the new model as the loss. During this phase, the outputs of the initially trained model are used as labels.

## B. Details on integrating our method into bound propagation

Bound propagation is an efficient framework for finding valid linear lower bounds for $c^T f(x)$ within some input set $x \in \mathcal{X}$. In this section, we first give a brief overview of performing bound propagation using the elementwise bound on preactivations. We then demonstrate how our results can be integrated into this framework to yield SDP-CROWN, an extension capable of performing bound propagation using the $\ell_2$-norm constraint on the preactivations.

We include a concrete example in Section B.3 to illustrate the details of both bound propagation and SDP-CROWN.

### B.1. LiRPA: bound propagation using the elementwise bound on preactivations

To better concept of bound propagation, we express each $z^{(k)}$ as a function of the input $x$, and redefine the neural network $f(x)$ in (1) as

$$f(x) = z^{(N)}(x), \quad z^{(k+1)}(x) = W^{(k+1)} \operatorname{ReLU}(z^{(k)}(x)), \quad z^{(1)}(x) = W^{(1)}x \quad \text{for } k = 1 \ldots, N-1.$$

The bound propagation we describe in this section is the backward mode of Linear Relaxation based Perturbation Analysis (LiRPA) in (Xu et al., 2020) that efficiently constructs linear lower bounds for $c^T f(x)$ within an elementwise bound relaxation of the input set $\mathcal{X}$:

$$g(\alpha)^T x + h(\alpha) \le c^T f(x) \text{ for all } x \in \mathcal{B}_\infty(\tilde{x}, \tilde{\rho}) \supseteq \mathcal{X}, \tag{9}$$

where $\tilde{x}, \tilde{\rho} \in \mathbb{R}^n$ are the radius and center of the elementwise bound.

LiRPA computes (9) by backward constructing linear lower bounds with respect to each preactivation $z^{(k)}(x)$

$$[g^{(k)}(\alpha^{(k)})]^T z^{(k)}(x) + h^{(k)}(\alpha^{(k)}) \le c^T f(x) \text{ for all } x \in \mathcal{B}_\infty(\tilde{x}, \tilde{\rho}), \tag{10}$$

i.e., starting from $k = N$ all the way down to $k = 1$. Here, $0 \le \alpha^{(k)} \le 1$ are variables of the same length as $z^{(k)}(x)$, which are defined in (11).

For $k = N$. The linear lower bound with respect to $z^{(k)}$ is simply

$$c^T z^{(k)}(x) \le c^T f(x),$$

and therefore, $g^{(k)}(\alpha^{(k)}) = c$ and $h^{(k)}(\alpha^{(k)}) = 0$.

For $k = N-1, \ldots, 1$. LiRPA takes linear lower bounds with respect to $z^{(k+1)}(x)$ ((10) at $k+1$) to construct linear bounds with respect to $z^{(k)}(x)$ ((10) at $k$). The first step is to substitute $z^{(k+1)}(x) = W^{(k+1)} \operatorname{ReLU}(z^{(k)}(x))$ to yield

$$[c^{(k)}]^T \operatorname{ReLU}(z^{(k)}(x)) + d^{(k)} \le c^T f(x)$$

where $c^{(k)} = [W^{(k+1)}]^T g^{(k+1)}(\alpha^{(k+1)})$ and $d^{(k)} = h^{(k+1)}(\alpha^{(k+1)})$. Since $\operatorname{ReLU}(z^{(k)}(x))$ is nonlinear, LiRPA performs linear relaxation on each $\operatorname{ReLU}(z_i^{(k)}(x))$ to propagate linear bounds from $\operatorname{ReLU}(z^{(k)}(x))$ to $z^{(k)}(x)$. In particular, LiRPA constructs linear bounds $\alpha_i^{(k)} z_i^{(k)}(x) \le \operatorname{ReLU}(z_i^{(k)}(x)) \le \beta_i^{(k)} z_i^{(k)}(x) + \gamma_i^{(k)}$ within $|z_i^{(k)}(x) - \tilde{z}_i^{(k)}| \le \tilde{\rho}_i^{(k)}$ where

$$\begin{cases} \alpha_i^{(k)} = 1, \quad \beta_i^{(k)} = 1, \quad \gamma_i^{(k)} = 0 & \text{if } \tilde{z}_i^{(k)} - \tilde{\rho}_i^{(k)} \ge 0 \\ \alpha_i^{(k)} = 0, \quad \beta_i^{(k)} = 0, \quad \gamma_i^{(k)} = 0 & \text{if } \tilde{z}_i^{(k)} + \tilde{\rho}_i^{(k)} \le 0 \\ \alpha_i^{(k)} = \tilde{\alpha}_i^{(k)}, \quad \beta_i^{(k)} = \frac{\tilde{z}_i^{(k)} + \tilde{\rho}_i^{(k)}}{2\tilde{\rho}_i^{(k)}}, \quad \gamma_i^{(k)} = -\frac{(\tilde{z}_i^{(k)} + \tilde{\rho}_i^{(k)})(\tilde{z}_i^{(k)} - \tilde{\rho}_i^{(k)})}{2\tilde{\rho}_i^{(k)}} & \text{otherwise,} \end{cases} \tag{11}$$

and $0 \le \tilde{\alpha}_i^{(k)} \le 1$ is a free variable that can be optimized (see Figure 5 for illustration). Here, the elementwise bound on each preactivation $|z_i^{(k)}(x) - \tilde{z}_i^{(k)}| \le \tilde{\rho}_i^{(k)}$ can be computed by treating $z_i^{(k)}(x)$ as the output of LiRPA, i.e., $c \equiv e_i$ and $f(x) \equiv z^{(k)}(x)$.

Using (11), the linear lower bounds with respect to $z^{(k)}$ are given by

$$\underbrace{[c_+^{(k)} \odot \alpha^{(k)} + c_-^{(k)} \odot \beta^{(k)}]}_{g^{(k)}(\alpha^{(k)})}{}^T z^{(k)}(x) + \underbrace{[c_-^{(k)}]^T \gamma^{(k)} + d^{(k)}}_{h^{(k)}(\alpha^{(k)})} \le [c^{(k)}]^T \operatorname{ReLU}(z^{(k)}(x)) + d^{(k)} \le c^T f(x) \tag{12}$$

where $c_+^{(k)} = \max\{c^{(k)}, 0\}$ and $c_-^{(k)} = \min\{c^{(k)}, 0\}$.

Finally, setting $g(\alpha) = [W^{(1)}]^T g^{(1)}(\alpha^{(1)})$ and $h(\alpha) = h^{(1)}(\alpha^{(1)})$ yields the desired linear lower bound in (9).

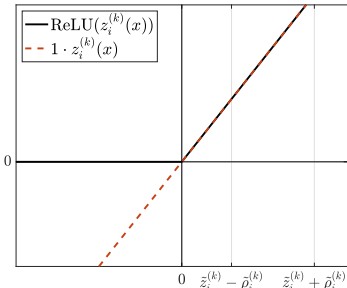 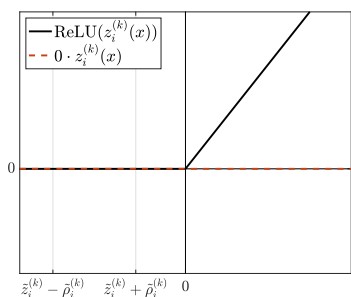 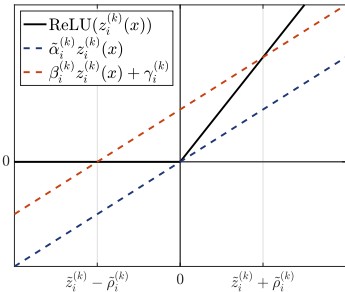

Figure 5: **Illustration of the linear relaxation (11)**. **(Left.)** $\tilde{z}_i^{(k)} - \tilde{\rho}_i^{(k)} \geq 0$. In this case, $\mathrm{ReLU}(z_i^{(k)}(x))$ is simply upper and lower bounded by $z_i^{(k)}(x)$. **(Middle.)** $\tilde{z}_i^{(k)} + \tilde{\rho}_i^{(k)} \leq 0$. In this case, $\mathrm{ReLU}(z_i^{(k)}(x))$ is simply upper and lower bounded by 0. **(Right.)** $\tilde{z}_i^{(k)} - \tilde{\rho}_i^{(k)} \leq 0 \leq \tilde{z}_i^{(k)} + \tilde{\rho}_i^{(k)}$. In this case, $\mathrm{ReLU}(z_i^{(k)}(x))$ is lower bounded by $\tilde{\alpha}_i^{(k)} z_i^{(k)}(x)$ for any $0 \leq \tilde{\alpha}_i^{(k)} \leq 1$ and upper bounded by a linear function intersects $(\tilde{z}_i^{(k)} - \tilde{\rho}_i^{(k)}, 0)$ and $(\tilde{z}_i^{(k)} + \tilde{\rho}_i^{(k)}, \tilde{z}_i^{(k)} + \tilde{\rho}_i^{(k)})$.

### B.2. SDP-CROWN: bound propagation using the $\ell_2$-norm constraint on preactivations

SDP-CROWN efficiently constructs linear lower bounds for $c^T f(x)$ within an $\ell_2$-norm ball relaxation of the input set $\mathcal{X}$:

$$g(\alpha)^T x + h(g(\alpha), \lambda) \leq c^T f(x) \ \text{ for all } \ x \in \mathcal{B}_2(\hat{x}, \rho) \supseteq \mathcal{X}, \tag{13}$$

where $\hat{x} \in \mathbb{R}^n$ and $\rho \in \mathbb{R}$ are the center and radius of the $\ell_2$-norm ball. To extend LiRPA to compute (13), we simply set $\tilde{x} = \hat{x}$, $\tilde{\rho} = \rho \mathbf{1}$ in (9) and follow the same process of LiRPA, except with the offsets $h^{(k)}(\alpha^{(k)})$ in (12) replaced by

$$h^{(k)}(g^{(k)}(\alpha^{(k)}), \lambda^{(k)}) = -\frac{1}{2} \cdot \left( \lambda^{(k)} \left( (\rho^{(k)})^2 - \|\hat{z}^{(k)}\|_2^2 \right) + \frac{1}{\lambda^{(k)}} \|\phi^{(k)}(g^{(k)}(\alpha^{(k)}), \lambda^{(k)})\|_2^2 \right) + d^{(k)}$$

where $\lambda^{(k)} \geq 0$ is a free variable that can be optimized and

$$\phi_i^{(k)}(g^{(k)}(\alpha^{(k)}), \lambda^{(k)}) = \min\{c_i^{(k)} - g_i^{(k)}(\alpha^{(k)}) - \lambda^{(k)}\hat{z}_i^{(k)}, \ g_i^{(k)}(\alpha^{(k)}) + \lambda^{(k)}\hat{z}_i^{(k)}, \ 0\} \quad \text{for all } \ i.$$

Here $\hat{z}^{(k)}$ and $\rho^{(k)}$ are the center and radius of the $\ell_2$ norm ball for $z^{(k)}(x)$, i.e., $\|z^{(k)}(x) - \hat{z}^{(k)}\|_2 \leq \rho^{(k)}$. The $\ell_2$ norm ball for $z^{(k)}(x)$ can be computed via the spectral norm of the weight matrices, as in $\hat{z}^{(k)} = W^{(k)}W^{(k-1)} \cdots W^{(1)}\hat{x}$ and $\rho^{(k)} = \|W^{(k)}\|_2 \|W^{(k-1)}\|_2 \cdots \|W^{(1)}\|_2 \rho$, or by more sophisticated methods such as (Fazlyab et al., 2019). Here, we use $\|W^{(k)}\|_2$ to denote the spectral $\ell_2$-norm of the matrix $W^{(k)}$. We note that by Theorem 4.1, we always have

$$[g^{(k)}(\alpha^{(k)})]^T z^{(k)}(x) + h^{(k)}(g^{(k)}(\alpha^{(k)}), \lambda^{(k)}) \leq c^T f(x) \ \text{ for all } \ x \in \mathcal{B}_2(\hat{x}, \rho) \tag{14}$$

for all $k = N, \ldots, 1$.

Finally, setting $g(\alpha) = [W^{(1)}]^T g^{(1)}(\alpha^{(1)})$ and $h(g(\alpha), \lambda) = h^{(1)}(g^{(1)}(\alpha^{(1)}), \lambda^{(1)})$ yields the desired linear lower bounds in (13).

### B.3. A small example of LiRPA and SDP-CROWN

We give a step-by-step illustration of how to find a linear lower bound on $c^T f(x)$ within $\|x - \hat{x}\|_2 \leq \rho$. Here, we consider a 3-layer neural network $f(x)$ with

$$c = 1, \quad W^{(3)} = \begin{bmatrix} -1 & -1 \end{bmatrix}, \quad W^{(2)} = \begin{bmatrix} -1 & 1 \\ 1 & -1 \end{bmatrix}, \quad W^{(1)} = \begin{bmatrix} 0 & 1 \\ 1 & 0 \end{bmatrix}, \quad \hat{x} = \begin{bmatrix} 1 \\ 1 \end{bmatrix}, \quad \rho = 1.$$

**LiRPA.** For simplicity, we compute each intermediate bound $|z^{(k)}(x) - \tilde{z}^{(k)}| \leq \tilde{\rho}^{(k)}$ via interval bound propagation (Gowal et al., 2019), and always pick $\tilde{\alpha}^{(k)} = 0$ in (11) for $k = 1, 2$. In this particular example, the choice of $\tilde{\alpha}^{(k)}$ does not affect the final result. The intermediate bounds are given by

$$\tilde{z}^{(1)} = \begin{bmatrix} 1 \\ 1 \end{bmatrix}, \quad \tilde{\rho}^{(1)} = \begin{bmatrix} 1 \\ 1 \end{bmatrix}, \quad \tilde{z}^{(2)} = \begin{bmatrix} 0 \\ 0 \end{bmatrix}, \quad \tilde{\rho}^{(2)} = \begin{bmatrix} 2 \\ 2 \end{bmatrix}.$$

- Starting at $k = 3$, we simply have

$$g^{(3)}(\alpha^{(3)}) = 1, \quad h^{(3)}(\alpha^{(3)}) = 0.$$

- At $k = 2$, substituting $z^{(3)}(x) = W^{(3)}\,\mathrm{ReLU}(z^{(2)}(x))$ and constructing $\alpha_i^{(2)} z_i^{(2)}(x) \leq \mathrm{ReLU}(z_i^{(2)}(x)) \leq \beta_i^{(2)} z_i^{(2)}(x) + \gamma_i^{(2)}$ via $|z^{(2)}(x) - \tilde{z}^{(2)}| \leq \tilde{\rho}^{(2)}$ gives

$$c^{(2)} = [W^{(3)}]^T g^{(3)}(\alpha^{(3)}) = \begin{bmatrix} -1 \\ -1 \end{bmatrix}, \quad d^{(2)} = h^{(3)}(\alpha^{(3)}) = 0, \quad \alpha^{(2)} = \begin{bmatrix} 0 \\ 0 \end{bmatrix}, \quad \beta^{(2)} = \begin{bmatrix} 0.5 \\ 0.5 \end{bmatrix}, \quad \gamma^{(2)} = \begin{bmatrix} 1 \\ 1 \end{bmatrix}.$$

Therefore, we have

$$g^{(2)}(\alpha^{(2)}) = c_+^{(2)} \odot \alpha^{(2)} + c_-^{(2)} \odot \beta^{(2)} = \begin{bmatrix} -0.5 \\ -0.5 \end{bmatrix}, \quad h^{(2)}(\alpha^{(2)}) = [c_-^{(2)}]^T \gamma^{(2)} + d^{(2)} = -2.$$

- At $k = 1$, substituting $z^{(2)}(x) = W^{(2)}\,\mathrm{ReLU}(z^{(1)}(x))$ and constructing $\alpha_i^{(1)} z_i^{(1)}(x) \leq \mathrm{ReLU}(z_i^{(1)}(x)) \leq \beta_i^{(1)} z_i^{(1)}(x) + \gamma_i^{(1)}$ via $|z^{(1)}(x) - \tilde{z}^{(1)}| \leq \tilde{\rho}^{(1)}$ gives

$$c^{(1)} = [W^{(2)}]^T g^{(2)}(\alpha^{(2)}) = \begin{bmatrix} 0 \\ 0 \end{bmatrix}, \quad d^{(1)} = h^{(2)}(\alpha^{(2)}) = -2, \quad \alpha^{(1)} = \begin{bmatrix} 1 \\ 1 \end{bmatrix}, \quad \beta^{(1)} = \begin{bmatrix} 1 \\ 1 \end{bmatrix}, \quad \gamma^{(1)} = \begin{bmatrix} 0 \\ 0 \end{bmatrix}.$$

Therefore, we have

$$g^{(1)}(\alpha^{(1)}) = c_+^{(1)} \odot \alpha^{(1)} + c_-^{(1)} \odot \beta^{(1)} = \begin{bmatrix} 0 \\ 0 \end{bmatrix}, \quad h^{(1)}(\alpha^{(1)}) = [c_-^{(1)}]^T \gamma^{(1)} + d^{(1)} = -2.$$

As a result, from LiRPA, we conclude

$$-2 = \begin{bmatrix} 0 \\ 0 \end{bmatrix}^T x + -2 \leq 1 \cdot f(x) \quad \text{for all} \quad \left\| x - \begin{bmatrix} 1 \\ 1 \end{bmatrix} \right\|_2 \leq 1.$$

**SDP-CROWN.** For simplicity, we compute each intermediate bound $\|z^{(k)}(x) - \hat{z}^{(k)}\|_2 \leq \rho^{(k)}$ for $k = 1, 2$ via the Lipschitz constant of $W^{(k)}$, which is given by

$$\hat{z}^{(1)} = \begin{bmatrix} 1 \\ 1 \end{bmatrix}, \quad \rho^{(1)} = 1, \quad \hat{z}^{(2)} = \begin{bmatrix} 0 \\ 0 \end{bmatrix}, \quad \rho^{(2)} = 2.$$

- Starting at $k = 3$, we simply have

$$g^{(3)}(\alpha^{(3)}) = 1, \quad h^{(3)}(g^{(3)}(\alpha^{(3)}), \lambda^{(3)}) = 0.$$

- At $k = 2$, taking $c^{(2)}, g^{(2)}(\alpha^{(2)})$ from LiRPA, setting $d^{(2)} = h^{(3)}(g^{(3)}(\alpha^{(3)}), \lambda^{(3)}) = 0$, and substituting them into

$$h^{(2)}(g^{(2)}(\alpha^{(2)}), \lambda^{(2)}) = -\frac{1}{2} \cdot \left( \lambda^{(2)} \left( (\rho^{(2)})^2 - \|\hat{z}^{(2)}\|_2^2 \right) + \frac{1}{\lambda^{(2)}} \|\phi^{(2)}(g^{(2)}(\alpha^{(2)}), \lambda^{(2)})\|_2^2 \right) + d^{(2)}$$

$$= -\frac{1}{2} \cdot \left( 4\lambda^{(2)} + \frac{1}{\lambda^{(2)}} \cdot 0.5 \right).$$

Notice that $\max_{\lambda^{(2)} \geq 0} h^{(2)}(g^{(2)}(\alpha^{(2)}), \lambda^{(2)})$ is maximized at $\lambda^{(2)} = \sqrt{1/8}$, and hence we have

$$g^{(2)}(\alpha^{(2)}) = \begin{bmatrix} -0.5 \\ -0.5 \end{bmatrix}, \quad h^{(2)}(g^{(2)}(\alpha^{(2)}), \lambda^{(2)}) = -\sqrt{2}, \quad \lambda^{(2)} = \sqrt{1/8}.$$

- At $k = 1$, taking $c^{(1)}$, $g^{(1)}(\alpha^{(1)})$ from LiRPA, setting $d^{(1)} = h^{(2)}(g^{(2)}(\alpha^{(2)}), \lambda^{(2)}) = -\sqrt{2}$, and substituting them into

$$
\begin{aligned}
h^{(1)}(g^{(1)}(\alpha^{(1)}), \lambda^{(1)}) &= -\frac{1}{2} \cdot \left( \lambda^{(1)} \left( (\rho^{(1)})^2 - \|\hat{z}^{(1)}\|_2^2 \right) + \frac{1}{\lambda^{(1)}} \|\phi^{(1)}(g^{(1)}(\alpha^{(1)}), \lambda^{(1)})\|_2^2 \right) + d^{(1)} \\
&= -\frac{1}{2} \cdot \left( -\lambda^{(1)} + \frac{1}{\lambda^{(1)}} \cdot \|\min\{-\lambda^{(1)}\hat{z}^{(1)}, \lambda^{(1)}\hat{z}^{(1)}, 0\}\|_2^2 \right) - \sqrt{2} \\
&= -\frac{1}{2} \cdot \left( -\lambda^{(1)} + \lambda^{(1)} \cdot \|\min\{-\hat{z}^{(1)}, \hat{z}^{(1)}, 0\}\|_2^2 \right) - \sqrt{2} \\
&= -\frac{1}{2} \cdot \left( \lambda^{(1)} \right) - \sqrt{2}.
\end{aligned}
$$

Obviously, $\max_{\lambda^{(1)} \geq 0} h^{(1)}(g^{(1)}(\alpha^{(1)}), \lambda^{(1)})$ is maximized at $\lambda^{(1)} = 0$, and hence

$$
g^{(1)}(\alpha^{(1)}) = \begin{bmatrix} 0 \\ 0 \end{bmatrix}, \quad h^{(1)}(g^{(1)}(\alpha^{(1)}), \lambda^{(1)}) = -\sqrt{2}, \quad \lambda^{(1)} = 0.
$$

As a result, from our method, we conclude

$$
-\sqrt{2} = \begin{bmatrix} 0 \\ 0 \end{bmatrix}^T x + -\sqrt{2} \leq 1 \cdot f(x) \text{ for all } \left\| x - \begin{bmatrix} 1 \\ 1 \end{bmatrix} \right\|_2 \leq 1.
$$

In this particular example, our method tightens bound propagation by exactly a factor of $\sqrt{2}$.

## C. Some extensions of SDP-CROWN

In this section, we describe several extensions that can further tighten SDP-CROWN.

### C.1. Ellipsoid constraints

The tightness of SDP-CROWN hinges crucially on the quality of the $\ell_2$-norm ball relaxation $\mathcal{B}_2(\hat{z}^{(k)}, \rho^{(k)}) \supseteq \{z^{(k)} \mid x \in \mathcal{B}_2(\hat{x}, \rho)\}$ for the input set at each $z^{(k)}$ during the computation of linear lower bounds (14). However, in general settings, $\ell_2$-norm balls might not be the best choice to relax the input set at $z^{(k)}$. For illustration, consider a simple one-layer example with

$$
W^{(1)} = \begin{bmatrix} 0.5 & 0.5 \\ 1.5 & -0.5 \end{bmatrix}, \quad \hat{x} = \begin{bmatrix} 0 \\ 0 \end{bmatrix}, \quad \rho = 1.
$$

As illustrated in Figure 6, the input set at $z^{(1)}$, $\{z^{(1)} \mid x \in \mathcal{B}_2(\hat{x}, \rho)\}$, is a rotated and elongated ellipsoid; therefore, relaxing this input set by naively propagating $\ell_2$-norm ball from $x$ to $z^{(1)}$ can result in extremely loose relaxation. To address this issue, we generalize SDP-CROWN to handle ellipsoids of the following form

$$
\mathcal{E}_2(\hat{x}, \hat{\rho}) = \{x \mid \|\operatorname{diag}(\hat{\rho})^{-1}(x - \hat{x})\|_2 \leq 1\}
$$

where $\hat{x}, \hat{\rho} \in \mathbb{R}^n$ are the center and axes of the ellipsoid.

We note that the ellipsoid can also be efficiently propagated via the Lipschitz constant of $W^{(k)}$. Given $\mathcal{E}_2(\hat{z}^{(k)}, \hat{\rho}^{(k)})$, one simple heuristic is to select the center and axis for $\mathcal{E}_2(\hat{z}^{(k+1)}, \hat{\rho}^{(k+1)})$ as

$$
\hat{z}^{(k+1)} = W^{(k+1)}\hat{z}^{(k)}, \quad \hat{\rho}^{(k+1)} = y \cdot \|\operatorname{diag}(y)^{-1} W^{(k+1)} \operatorname{diag}(\hat{\rho}^{(k)})\|_2 \text{ with } y = \|W^{(k+1)} \operatorname{diag}(\hat{\rho}^{(k)})\|_{r,2}
$$

where $\|W\|_{r,2} = \sqrt{(W \odot W)\mathbf{1}}$ denotes the rowwise $\ell_2$ norm of a matrix $W$.

### C.2. Intersection between ellipsoid and elementwise constraints

Another extension of SDP-CROWN is to also take the elementwise bound $\mathcal{B}_\infty(\tilde{z}^{(k)}, \tilde{\rho}^{(k)})$ at $z^{(k)}$ into account when computing the relaxation of (7). In particular, SDP-CROWN can be further tightened by considering the intersection between the ellipsoid $\mathcal{E}_2(\hat{z}^{(k)}, \hat{\rho}^{(k)})$ and the elementwise bound $\mathcal{B}_\infty(\tilde{z}^{(k)}, \tilde{\rho}^{(k)})$ at $z^{(k)}$. For illustration, in Figure 7, we plot the

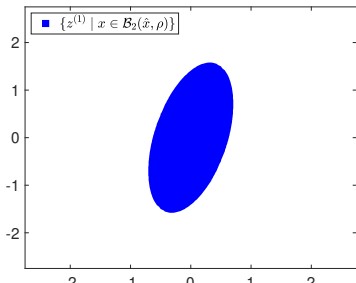 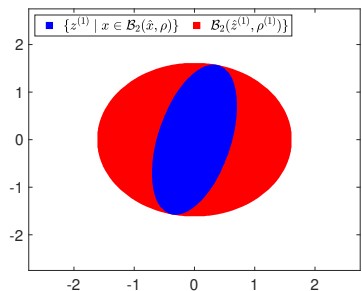 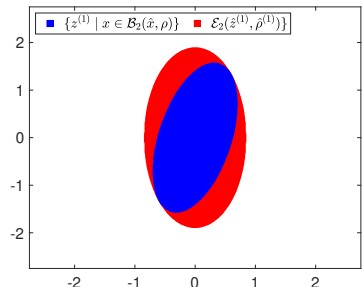

Figure 6: Constructing the $\ell_2$-norm ball and the ellipsoid relaxation at $z^{(1)}$ for a one layer neural network with $W^{(1)} = [0.5, 0.5; 1.5, -0.5]$, $\hat{x} = [0; 0]$ and $\rho = 1$. (**Left.**) The input set at $z^{(1)}$ with respect to the $\ell_2$-norm ball input set $\mathcal{B}_2(\hat{x}, \rho)$ at $x$. The input set at $z^{(1)}$ is a rotated and elongated ellipsoid. (**Middle.**) The $\ell_2$-norm ball relaxation $\mathcal{B}_2(\hat{z}^{(1)}, \rho^{(1)})$ at $z^{(1)}$, where $\hat{z}^{(1)} = [0; 0]$ and $\rho^{(1)} = \|W^{(1)}\|_2 \rho = 1.5302$. $\ell_2$-norm ball does not have enough degree of freedom to capture the shape of the input set at $z^{(1)}$. (**Right.**) The ellipsoid relaxation $\mathcal{E}_2(\hat{z}_1, \hat{\rho}^{(1)})$ at $z^{(1)}$, where $\hat{z}^{(1)} = [0; 0]$ and $\hat{\rho}^{(1)} = [0.5464; 1.9360]$. The ellipsoid can better capture the shape of the input set at $z^{(1)}$.

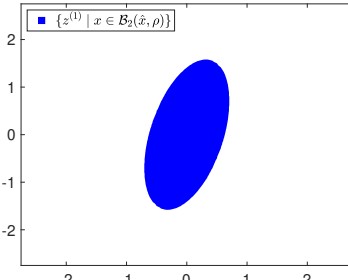 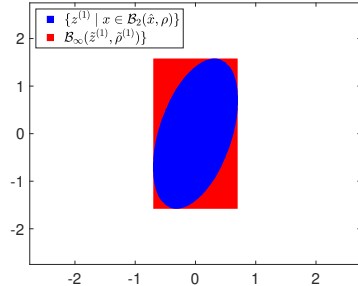 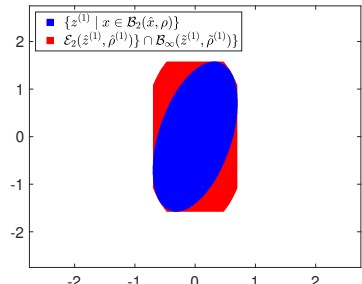

Figure 7: Constructing the intersection between ellipsoid and elementwise bound as a relaxation at $z^{(1)}$ for a one layer neural network with $W^{(1)} = [0.5, 0.5; 1.5, -0.5]$, $\hat{x} = [0; 0]$ and $\rho = 1$. (**Left.**) The input set at $z^{(1)}$ with respect to the $\ell_2$-norm ball input set $\mathcal{B}_2(\hat{x}, \rho)$ at $x$. The input set at $z^{(1)}$ is a rotated and elongated ellipsoid. (**Middle.**) The elementwise bound relaxation $\mathcal{B}_\infty(\tilde{z}^{(1)}, \tilde{\rho}^{(1)})$ at $z^{(1)}$, where $\tilde{z}^{(1)} = [0; 0]$ and $\tilde{\rho}^{(1)} = \|W^{(1)}\|_{r,2} \rho = [0.7071; 1.5811]$. (**Right.**) The intersection between ellipsoid $\mathcal{E}_2(\hat{z}_1, \hat{\rho}^{(1)})$ and elementwise bound $\mathcal{B}_\infty(\tilde{z}^{(1)}, \tilde{\rho}^{(1)})$ at $z^{(1)}$, where $\hat{z}^{(1)} = [0; 0]$ and $\hat{\rho}^{(1)} = [0.5464; 1.9360]$. The intersection removes the corners of $\mathcal{B}_\infty(\tilde{z}^{(1)}, \tilde{\rho}^{(1)})$.

intersection between the elementwise bound $\mathcal{B}_\infty(\tilde{z}^{(k)}, \tilde{\rho}^{(k)})$ and the ellipsoid $\mathcal{E}_2(\hat{z}^{(k)}, \hat{\rho}^{(k)})$ using the same one-layer example in Figure 6. In this case, the intersection removes four corners of $\mathcal{B}_\infty(\tilde{z}^{(k)}, \tilde{\rho}^{(k)})$. We note that as the dimension increases, the number of corners removed grows exponentially.

We can simply accommodate $\mathcal{B}_\infty(\tilde{z}^{(k)}, \tilde{\rho}^{(k)})$ by adding the following inequality constraint into (7)

$$\mathrm{ReLU}(x_i) \leq \beta_i^{(k)} x + \gamma_i^{(k)}$$

where $\beta_i^{(k)}$ and $\gamma_i^{(k)}$ are defined in (11). We summarized the extension of SDP-CROWN for handling the intersection between the elementwise bound $\mathcal{B}_\infty(\tilde{z}^{(k)}, \tilde{\rho}^{(k)})$ and the ellipsoid $\mathcal{E}_2(\hat{z}^{(k)}, \hat{\rho}^{(k)})$ in the following Theorem.

**Theorem C.1.** *Given* $c, \hat{x}, \hat{\rho}, \tilde{x}, \tilde{\rho} \in \mathbb{R}^n$ *where* $\hat{\rho}, \tilde{\rho} \geq 0$. *The following holds*

$$c^T \mathrm{ReLU}(x) \geq g^T x + h(g, \lambda, \tau) \ \text{for all} \ x \in \mathcal{E}_2(\hat{x}, \hat{\rho}) \cap \mathcal{B}_\infty(\tilde{x}, \tilde{\rho})$$

*for any* $\lambda, \tau \geq 0$ *and* $g \in \mathbb{R}^n$ *where*

$$h(g, \lambda, \tau) = -\frac{1}{2} \left( \lambda(1 - \| \mathrm{diag}(\hat{\rho})^{-1} \hat{x} \|_2^2) + 2\tau^T (\tilde{\rho} \odot \tilde{\rho} - \tilde{x} \odot \tilde{x}) + \frac{1}{\lambda} \|\phi(g, \lambda, \tau)\|_2^2 \right)$$

*and*

$$\phi_i(g, \lambda, \tau) = \hat{\rho}_i \cdot \min\{c_i - g_i + \tau_i(\tilde{\rho}_i - \tilde{x}_i) - \lambda\hat{\rho}_i^{-2}\hat{x}_i, g_i + \tau_i(\tilde{\rho}_i + \tilde{x}_i) + \lambda\hat{\rho}_i^{-2}\hat{x}_i, 0\}.$$

## C.3. Proof of Theorem C.1

Given a linear relaxation $c^T \operatorname{ReLU}(x) \geq g^T x + h$ that holds within $x \in \mathcal{E}_2(\hat{x}, \hat{\rho}) \cap \mathcal{B}_\infty(\tilde{x}, \tilde{\rho})$, the process of finding the tightest possible $h$ within $\mathcal{E}_2(\hat{x}, \hat{\rho}) \cap \mathcal{B}_\infty(\tilde{x}, \tilde{\rho})$ admits the following generic problem

$$\min_{x \in \mathbb{R}^n} c^T \operatorname{ReLU}(x) - g^T x \quad \text{s.t.} \quad \|\operatorname{diag}(\hat{\rho})^{-1}(x - \hat{x})\|_2 \leq 1, \quad \operatorname{ReLU}(x_i) \leq \frac{\tilde{\rho}_i + \tilde{x}_i}{2\tilde{\rho}_i} x + \frac{\tilde{\rho}_i^2 - \tilde{x}_i^2}{2\tilde{\rho}_i} \quad \text{for } i = 1, \ldots, n.$$

Without loss of generality, we assume $\tilde{x}_i - \tilde{\rho}_i \leq 0 \leq \tilde{x}_i + \tilde{\rho}_i$ for all $i$. Applying the positive/negative splitting $x = u - v$ where $u, v \geq 0$ and $u \odot v = 0$ yields the following

$$\min_{u,v \in \mathbb{R}^n} c^T u - g^T(u - v)$$
$$\text{s.t.} \quad \sum_{i=1}^n (\hat{\rho}_i^{-1} u_i)^2 - 2\hat{\rho}_i^{-2}(u_i - v_i)\hat{x}_i + (\hat{\rho}_i^{-1} v_i)^2 \leq 1 - \|\operatorname{diag}(\hat{\rho})^{-1}\hat{x}\|_2^2, \tag{15}$$
$$(\tilde{\rho}_i - \tilde{x}_i)u_i + (\tilde{\rho}_i + \tilde{x}_i)v_i \leq \tilde{\rho}_i^2 - \tilde{x}_i^2 \quad \text{for } i = 1, \ldots, n,$$
$$u \geq 0, \quad v \geq 0, \quad u \odot v = 0.$$

The SDP relaxation of (15) reads:

$$\min_{\tilde{u}, \tilde{v}, u, v, U, V \in \mathbb{R}^n} c^T u - g^T(u - v)$$
$$\text{s.t.} \quad \sum_{i=1}^n \hat{\rho}_i^{-2} U_i - 2\hat{\rho}_i^{-2}(u_i - v_i)\hat{x}_i + \hat{\rho}_i^{-2} V_i \leq 1 - \|\operatorname{diag}(\hat{\rho})^{-1}\hat{x}\|_2^2,$$
$$(\tilde{\rho}_i - \tilde{x}_i)u_i + (\tilde{\rho}_i + \tilde{x}_i)v_i \leq \tilde{\rho}_i^2 - \tilde{x}_i^2 \quad \text{for } i = 1, \ldots, n,$$
$$u \geq 0, \quad v \geq 0, \quad \tilde{u} + \tilde{v} = 1,$$
$$\begin{bmatrix} \tilde{u}_i & u_i \\ u_i & U_i \end{bmatrix} \succeq 0, \quad \begin{bmatrix} \tilde{v}_i & v_i \\ v_i & V_i \end{bmatrix} \succeq 0 \quad \text{for } i = 1, \ldots, n.$$

Let $\lambda \in \mathbb{R}$ denote the dual variable of the first inequality constraints, $\tau_i \in \mathbb{R}$ denote the dual variable of each $(\tilde{\rho}_i - \tilde{x}_i)u_i + (\tilde{\rho}_i + \tilde{x}_i)v_i \leq \tilde{\rho}_i^2 - \tilde{x}_i^2$, and $s, t, \mu \in \mathbb{R}^n$ denote the dual variable for $u \geq 0$, $v \geq 0$ and $\tilde{u} + \tilde{v} = 1$, respectively. The Lagrangian dual is given by

$$\max_{\lambda, \tau, s, t, \mu} -\frac{1}{2} \cdot \left( \lambda(1 - \|\operatorname{diag}(\hat{\rho})^{-1}\hat{x}\|_2^2) + 2\tau^T(\tilde{\rho} \odot \tilde{\rho} - \tilde{x} \odot \tilde{x}) + \mu^T \mathbf{1} \right)$$
$$\text{s.t.} \quad \begin{bmatrix} \mu_i & c_i - g_i + \tau_i(\tilde{\rho}_i - \tilde{x}_i) - \lambda\hat{\rho}_i^{-2}\hat{x}_i - s_i \\ c_i - g_i + \tau_i(\tilde{\rho}_i - \tilde{x}_i) - \lambda\hat{\rho}_i^{-2}\hat{x}_i - s_i & \hat{\rho}_i^{-2}\lambda \end{bmatrix} \succeq 0 \quad \text{for } i = 1, \ldots, n,$$
$$\begin{bmatrix} \mu_i & g_i + \tau_i(\tilde{\rho}_i + \tilde{x}_i) + \lambda\hat{\rho}_i^{-2}\hat{x}_i - t_i \\ g_i + \tau_i(\tilde{\rho}_i + \tilde{x}_i) + \lambda\hat{\rho}_i^{-2}\hat{x}_i - t_i & \hat{\rho}_i^{-2}\lambda \end{bmatrix} \succeq 0 \quad \text{for } i = 1, \ldots, n,$$
$$\lambda \geq 0, \quad \tau \geq 0, \quad s \geq 0, \quad t \geq 0, \quad \mu \geq 0.$$

For a $2 \times 2$ matrix, note that $X \succeq 0$ holds if and only if $\det(X) \geq 0$ and $\operatorname{diag}(X) \geq 0$. Applying this insight yields a second-order cone programming (SOCP) problem

$$\max_{\lambda, \tau, s, t, \mu} -\frac{1}{2} \cdot \left( \lambda(1 - \|\operatorname{diag}(\hat{\rho})^{-1}\hat{x}\|_2^2) + 2\tau^T(\tilde{\rho} \odot \tilde{\rho} - \tilde{x} \odot \tilde{x}) + \mu^T \mathbf{1} \right)$$
$$\text{s.t.} \quad \mu_i \lambda \geq \hat{\rho}_i^2 (c_i - g_i + \tau_i(\tilde{\rho}_i - \tilde{x}_i) - \lambda\hat{\rho}_i^{-2}\hat{x}_i - s_i)^2, \tag{16}$$
$$\mu_i \lambda \geq \hat{\rho}_i^2 (g_i + \tau_i(\tilde{\rho}_i + \tilde{x}_i) + \lambda\hat{\rho}_i^{-2}\hat{x}_i - t_i)^2,$$
$$\lambda \geq 0, \quad \tau \geq 0, \quad s \geq 0, \quad t \geq 0, \quad \mu \geq 0.$$

We are now ready to prove Theorem C.1.

*Proof.* Given any $c, g \in \mathbb{R}^n$. Let $a_i = \hat{\rho}_i(c_i - g_i + \tau_i(\tilde{\rho}_i - \tilde{x}_i) - \lambda\hat{\rho}_i^{-2}\hat{x}_i)$ and $b_i = \hat{\rho}_i(g_i + \tau_i(\tilde{\rho}_i + \tilde{x}_i) + \lambda\hat{\rho}_i^{-2}\hat{x}_i)$. Fixing any $\lambda, \tau \geq 0$ and optimizing $\mu$ in (16) yields

$$
\max_{\lambda, s, t \geq 0} -\frac{1}{2} \cdot \left( \lambda(1 - \|\operatorname{diag}(\hat{\rho})^{-1}\hat{x}\|_2^2) + 2\tau^T(\tilde{\rho} \odot \tilde{\rho} - \tilde{x} \odot \tilde{x}) + \sum_{i=1}^{n} \frac{\max\left\{(a_i - s_i)^2, (b_i - t_i)^2\right\}}{\lambda} \right)
$$

$$
= \max_{\lambda \geq 0} -\frac{1}{2} \cdot \left( \lambda(1 - \|\operatorname{diag}(\hat{\rho})^{-1}\hat{x}\|_2^2) + 2\tau^T(\tilde{\rho} \odot \tilde{\rho} - \tilde{x} \odot \tilde{x}) + \sum_{i=1}^{n} \frac{\min\left\{a_i, b_i, 0\right\}^2}{\lambda} \right)
$$

$$
= \max_{\lambda \geq 0} h(g, \lambda, \tau)
$$

where the first equality follows from $\min_{s_i \geq 0}(a_i - s_i)^2 = \min\{a_i, 0\}^2$ and $\min_{t_i \geq 0}(b_i - t_i)^2 = \min\{b_i, 0\}^2$, and $\max\{\min\{a_i, 0\}^2, \min\{b_i, 0\}^2\} = \min\{a_i, b_i, 0\}^2$ for any $a_i, b_i \in \mathbb{R}$. Since $h(g, \lambda, \tau)$ is a lower bound on (15) for any $\lambda, \tau \geq 0$, we have $c^T \operatorname{ReLU}(x) \geq g^T x + h(g, \lambda, \tau)$ for all $x \in \mathcal{E}_2(\hat{x}, \hat{\rho}) \cap \mathcal{B}_\infty(\tilde{x}, \tilde{\rho})$ for any $g \in \mathbb{R}^n$, $\lambda, \tau \geq 0$. $\qquad\square$

## D. Derivation of the dual problem (8)

Recall that we have the primal problem

$$
\min_{\tilde{u}, \tilde{v}, u, v, U, V \in \mathbb{R}^n} c^T u - g^T(u - v)
$$

$$
\text{s.t.} \quad (U + V)^T \mathbf{1} - 2(u - v)^T \hat{x} \leq \rho^2 - \|\hat{x}\|_2^2,
$$

$$
u \geq 0, \quad v \geq 0, \quad \tilde{u} + \tilde{v} = 1,
$$

$$
\begin{bmatrix} \tilde{u}_i & u_i \\ u_i & U_i \end{bmatrix} \succeq 0, \quad \begin{bmatrix} \tilde{v}_i & v_i \\ v_i & V_i \end{bmatrix} \succeq 0 \quad \text{for } i = 1, \ldots, n.
$$

Let $\lambda \geq 0$ denote the dual variables of the first inequality constraints. $s, t \geq 0$, $\mu \in \mathbb{R}^n$ denote the dual variable for $u \geq 0$, $v \geq 0$ and $\tilde{u} + \tilde{v} = 1$, respectively. $\begin{bmatrix} \tilde{y}_i & y_i \\ y_i & Y_i \end{bmatrix} \succeq 0$, $\begin{bmatrix} \tilde{z}_i & z_i \\ z_i & Z_i \end{bmatrix} \succeq 0$ denote the dual variables of the last two PSD constraints for $i = 1, \ldots n$. The Lagrangian is given by

$$
\mathcal{L}(\tilde{u}, \tilde{v}, u, v, U, V, \lambda, s, t, \mu, \tilde{y}, \tilde{z}, y, z, Y, Z) = \sum_{i=1}^{n} c_i u_i - g_i(u_i - v_i)
$$

$$
+ \left[ \sum_{i=1}^{n} \lambda(U_i + V_i) - 2\lambda\hat{x}_i(u_i - v_i) \right] - \lambda(\rho^2 - \|\hat{x}\|_2^2)
$$

$$
- \sum_{i=1}^{n}(s_i u_i + t_i v_i) + \sum_{i=1}^{n} \mu_i(\tilde{u}_i + \tilde{v}_i - 1)
$$

$$
- \sum_{i=1}^{n} \left\langle \begin{bmatrix} \tilde{y}_i & y_i \\ y_i & Y_i \end{bmatrix}, \begin{bmatrix} \tilde{u}_i & u_i \\ u_i & U_i \end{bmatrix} \right\rangle - \sum_{i=1}^{n} \left\langle \begin{bmatrix} \tilde{z}_i & z_i \\ z_i & Z_i \end{bmatrix}, \begin{bmatrix} \tilde{v}_i & v_i \\ v_i & V_i \end{bmatrix} \right\rangle.
$$

Rearranging the terms, we have

$$
\mathcal{L}(\tilde{u}, \tilde{v}, u, v, U, V, \lambda, s, t, \mu, \tilde{y}, \tilde{z}, y, z, Y, Z) = -\lambda(\rho^2 - \|\hat{x}\|_2^2) - \sum_{i=1}^{n} \mu_i
$$

$$
+ \sum_{i=1}^{n} \left\langle \begin{bmatrix} \mu_i - \tilde{y}_i & \frac{1}{2}(c_i - g_i - 2\lambda\hat{x}_i - s_i) - y_i \\ \frac{1}{2}(c_i - g_i - 2\lambda\hat{x}_i - s_i) - y_i & \lambda - Y_i \end{bmatrix}, \begin{bmatrix} \tilde{u}_i & u_i \\ u_i & U_i \end{bmatrix} \right\rangle \tag{17}
$$

$$
+ \sum_{i=1}^{n} \left\langle \begin{bmatrix} \mu_i - \tilde{z}_i & \frac{1}{2}(c_i + 2\lambda\hat{x}_i - t_i) - z_i \\ \frac{1}{2}(c_i + 2\lambda\hat{x}_i - t_i) - z_i & \lambda - Z_i \end{bmatrix}, \begin{bmatrix} \tilde{v}_i & v_i \\ v_i & V_i \end{bmatrix} \right\rangle. \tag{18}
$$

Minimizing the Lagrangian over the primal variables yields

$$\min_{\tilde{u},\tilde{v},u,v,U,V\in\mathbb{R}^n} \mathcal{L}(\tilde{u},\tilde{v},u,v,U,V,\lambda,s,t,\mu,\tilde{y},\tilde{z},y,z,Y,Z)$$

$$= \begin{cases} -\lambda(\rho^2 - \|\hat{x}\|_2^2) - \sum_{i=1}^n \mu_i & \text{if (17)} = 0 \text{ and (18)} = 0 \text{ for all } \tilde{u},\tilde{v},u,v,U,V \in \mathbb{R}^n \\ -\infty & \text{otherwise} \end{cases}$$

where

$$\text{(17)} = 0 \iff \begin{bmatrix} \tilde{y}_i & y_i \\ y_i & Y_i \end{bmatrix} = \begin{bmatrix} \mu_i & \frac{1}{2}(c_i - g_i - 2\lambda\hat{x}_i - s_i) \\ \frac{1}{2}(c_i - g_i - 2\lambda\hat{x}_i - s_i) & \lambda \end{bmatrix} \quad \text{for all } i \in \{1,\ldots,n\}$$

$$\text{(18)} = 0 \iff \begin{bmatrix} \tilde{z}_i & z_i \\ z_i & Z_i \end{bmatrix} = \begin{bmatrix} \mu_i & \frac{1}{2}(c_i + 2\lambda\hat{x}_i - t_i) \\ \frac{1}{2}(c_i + 2\lambda\hat{x}_i - t_i) & \lambda \end{bmatrix} \quad \text{for all } i \in \{1,\ldots,n\}.$$

Hence, the Lagrangian dual is given by

$$\max_{\lambda,s,t,\mu} \; -\lambda(\rho^2 - \|\hat{x}\|_2^2) - \mu^T \mathbf{1}$$

$$\text{s.t.} \quad \begin{bmatrix} \mu_i & \frac{1}{2}(c_i - g_i - 2\lambda\hat{x}_i - s_i) \\ \frac{1}{2}(c_i - g_i - 2\lambda\hat{x}_i - s_i) & \lambda \end{bmatrix} \succeq 0 \quad \text{for } i = 1,\ldots,n,$$

$$\begin{bmatrix} \mu_i & \frac{1}{2}(c_i + 2\lambda\hat{x}_i - t_i) \\ \frac{1}{2}(c_i + 2\lambda\hat{x}_i - t_i) & \lambda \end{bmatrix} \succeq 0 \quad \text{for } i = 1,\ldots,n,$$

$$\lambda \geq 0, \quad s \geq 0, \quad t \geq 0, \quad \mu \geq 0.$$

Rescaling $\lambda \equiv \frac{1}{2}\lambda$ and $\mu \equiv \frac{1}{2}\mu$ to yield

$$\max_{\lambda,s,t,\mu} \; -\frac{1}{2}\lambda(\rho^2 - \|\hat{x}\|_2^2) - \frac{1}{2}\mu^T \mathbf{1}$$

$$\text{s.t.} \quad \begin{bmatrix} \mu_i & c_i - g_i - \lambda\hat{x}_i - s_i \\ c_i - g_i - \lambda\hat{x}_i - s_i & \lambda \end{bmatrix} \succeq 0 \quad \text{for } i = 1,\ldots,n,$$

$$\begin{bmatrix} \mu_i & g_i + \lambda\hat{x}_i - t_i \\ g_i + \lambda\hat{x}_i - t_i & \lambda \end{bmatrix} \succeq 0 \quad \text{for } i = 1,\ldots,n,$$

$$\lambda \geq 0, \quad s \geq 0, \quad t \geq 0, \quad \mu \geq 0.$$

For a $2 \times 2$ matrix, note that $X \succeq 0$ holds if and only if $\det(X) \geq 0$ and $\text{diag}(X) \geq 0$. Finally, applying this insight yields the desired dual problem:

$$\frac{1}{2} \cdot \max_{\lambda,s,t,\mu} \; -\lambda(\rho^2 - \|\hat{x}\|_2^2) - \mu^T \mathbf{1}$$

$$\text{s.t.} \quad \lambda\mu_i \geq (c_i - g_i - s_i - \lambda\hat{x}_i)^2 \quad \text{for } i = 1,\ldots,n,$$

$$\lambda\mu_i \geq (-g_i + t_i - \lambda\hat{x}_i)^2 \quad \text{for } i = 1,\ldots,n,$$

$$\lambda \geq 0, \quad s \geq 0, \quad t \geq 0, \quad \mu \geq 0.$$

