# OpenReview forum: "SDP-CROWN: Efficient Bound Propagation for Neural Network Verification with Tightness of Semidefinite Programming"
_ICML.cc/2025/Conference — ICML 2025 spotlightposter_

### Official Review · Reviewer_zfFU · 2025-03-06

**Overall Recommendation:** 4

**Summary:**

The paper introduces SDP-CROWN, a hybrid framework combining semidefinite programming (SDP) relaxations with bound propagation for neural network verification under L2-norm perturbations. The core contribution is a novel linear bound derived from SDP principles that includes a new bias term h, which provides theoretical guarantees for L2-norm balls. Theoretical analysis shows the proposed bound can be tighter than bounds computed for the L-infinity norm. Experimental results demonstrate that SDP-CROWN outperforms state-of-the-art verifiers like α-CROWN and α,β-CROWN while maintaining moderate runtime.

**Claims And Evidence:**

Overall, the claims are supported by clear experimental results and theorem:
Tighter bounds under L2 perturbations: Empirical results (Table 1, Figure 3 and 4) show SDP-CROWN achieves significant verified accuracy improvement, outperforming baselines.
Scalability: Experiments on models with 65k neurons validate scalability.
Theoretical tightness: Theorem 5.2 proves that in a special yet crucial case, the proposed bound can achieve an improvement by a factor of the square root of n.

**Essential References Not Discussed:**

While I keep up with the literature on neural network verification, particularly the CROWN series, my understanding of SDP-based and Lipschitz-based verification is rather limited. Therefore, my main concern is whether more advanced methods exist, as the evaluation only considers a “naive” Lipschitz approach.

**Experimental Designs Or Analyses:**

I have reviewed the experimental setup, and overall, it is valid. The only concern is that I am unsure whether the chosen radius for L2-norm perturbation robustness evaluation is reasonable.

**Methods And Evaluation Criteria:**

The method is well-motivated for L2 robustness verification. Evaluation uses standard MNIST/CIFAR-10 benchmarks and models that are widely recognized in this field. It compares against relevant baselines (alpha-CROWN, alpha-beta-CROWN, LP-Full). The only aspect that may need further clarification is the baseline Naive_Lipschitz, which could benefit from a more detailed explanation.

**Other Comments Or Suggestions:**

Typo: line 171 “lose” -> “loose”.

**Other Strengths And Weaknesses:**

1. I appreciate Section 3 which clearly explains why and how the existing SOTA methods produce loose bounds under L2-norm perturbation.
2. Both the theoretical analysis and empirical results demonstrate that the proposed method represents a valuable step forward in addressing this problem.

**Questions For Authors:**

1. As I understand it, the core idea of this paper is that an L2-norm ball with the same radius is a subset of an L-infinity norm box, which allows existing bound propagation-based verification methods to be further tightened by focusing on L2-norm balls. The key improvement in this work lies in obtaining a tighter linear bias term h. My question is whether this idea is limited to refining the bias term h, or if it can also be extended to improve the slope term g for even tighter bounds.
2. In the evaluation on MLP models and the MNIST dataset, the baseline BM exhibits superior performance (even approaching the upper bound given by PGD). However, I could not find an introduction to BM—did I overlook something?
3. The selected baselines include alpha-CROWN and alpha-beta-CROWN. As I understand it, these existing CROWN-based tools directly verify L-infinity norm boxes rather than L2-norm balls under the current experimental setup. Would GCP-CROWN be inapplicable to this experiment? To my knowledge, GCP-CROWN has demonstrated stronger verification capabilities. Additionally, the Lipschitz-based baseline appears to exhibit a relatively favorable overhead. Are there more advanced methods in this category? A more detailed introduction to this baseline may also be necessary.
4. Again, I am not very familiar with SDP-based verification techniques, so I had some confusion while reading Lines 120–135. According to the authors, U_i = u_i^2 and V_i = v_i^2, and the constructed matrix X_i being positive semidefinite seems to directly imply u_i v_i = 0. This appears to be an exact equivalence to ReLU activation—where is the relaxation incorporated in this formulation? Based on my understanding, the conditions U_i \geq u_i and V_i \geq v_i seem to make more sense.

**Relation To Broader Scientific Literature:**

The work effectively bridges bound propagation-based verification and SDP technique. It aims to verify model robustness under L2-norm perturbation and thus related to neural network security.

**Theoretical Claims:**

I have reviewed the theorems in the main text and did not find any major issues.

---

> ### Author Rebuttal · Authors · 2025-03-31
>
> We want to thank reviewer zfFU for for your positive feedback and valuable comments. As suggested, we’ve added new experiments on GCP-CROWN [2] and BICCOS [3] (a follow-up work of GCP-CROWN), as well as a more sophisticated Lipschitz constant method (LipSDP [1]).
>
> **New experimental results**
>
> We provide additional comparisons with LipSDP [1], GCP-CROWN [2], and BICCOS [3] in the table below, using the same settings as Table 1 in our paper. We report the verified accuracy (%) and average per-sample verification time (in seconds), except for LipSDP, where we report the total time required to compute the Lipschitz constant.
>
> [Table A: Comparisons to new baselines](https://imgur.com/a/QITgjmq)
>
> Since GCP-CROWN and BICCOS are all part of the $\alpha,\beta$-CROWN verifier, to avoid confusion, we renamed the original $\alpha,\beta$-CROWN baseline to $\beta$-CROWN (Wang et al., 2021). GCP-CROWN and BICCOS marginally improve over $\beta$-CROWN, although the gap to our method is still significant, especially on large models. LipSDP can outperform native Lipschitz but is still not competitive compared to our method, and computing the Lipschitz constant using SDP is also quite slow and not scalable.
>
> [1] Fazlyab et al. "Efficient and accurate estimation of lipschitz constants for deep neural networks." NeurIPS 2019
>
> [2] Zhang et al. "General cutting planes for bound-propagation-based neural network verification." NeurIPS 2020
>
> [3] Zhou et al. "Scalable Neural Network Verification with Branch-and-bound Inferred Cutting Planes." NeurIPS 2024
>
> **"Whether the chosen radius for L2-norm perturbation is reasonable"**
>
> While Table 1 only presents results for a single $\ell_2$ perturbation radius, we would like to highlight Figure 4 (c) (d) in Appendix (also [linked here](https://imgur.com/a/MFUS3RX)), which presents the certified lower bounds computed from different methods over **a wide range of $\ell_2$ perturbation radii**. As shown in Figure 4, our lower is consistently tighter than $\alpha$-CROWN and the naive Lipschitz approach across different perturbation radii, which demonstrates the effectiveness of our approach.
>
> **"More advanced methods beyond naive Lipschitz"**
>
> We agree that more advanced methods based on Lipschitz constant estimation can yield significantly tighter robustness bounds. To address this, we have added a comparison with LipSDP [1] in the [Table A](https://imgur.com/a/QITgjmq) above, which employs SDP relaxation to estimate the network’s Lipschitz constant. This provides a stronger baseline for Lipschitz-based verification methods. Our method still significantly outperforms this baseline in all scenarios.
>
> **Response to Questions For Authors:**
> 1.  While our current work focuses on refining the bias term $h$, it is certainly possible to refine both the slope term $g$ and bias term $h$ simultaneously for tighter bounds. A key direction for future work is developing an efficient parameterization of $g$ that can be seamlessly integrated into existing bound propagation frameworks.
> 2. BM is an SDP-based method [4]. It exactly solves the SDP relaxation of the verification problem using specialized low-rank solvers. BM is one of the tightest SDP-based verifiers (as demonstrated in their paper) but is not scalable to CIFAR-10 models. We choose BM because it is a recently published SDP-based method with strong results.
>
> 3. We have added a comparison with GCP-CROWN [2] and BICCOS [3] above. The original GCP-CROWN implementation considers $\ell_\infty$ norm only, but we’ve able to extend the implementation to $\ell_2$ norm by finding cutting planes specifically for $\ell_2$ norm input constraints. It marginally improves performance and the gap between GCP-CROWN and our method is still large esepcially on bigger models.
> 4. We note that without $\mathrm{rank}(X_i)=1$, $X_i\succeq 0$ alone does not imply $u_iv_i = 0$, and the relaxation stems from dropping the constraint $\mathrm{rank}(X_i)=1$. Specifically, to derive SDP relaxation of ReLU activation: $x_i=u_i$, $u_iv_i=0$, $u_i\geq 0$, $v_i\geq 0$. We first add a redundant constraint $X_i=[1\ u_i\ v_i][1\ u_i\ v_i]^T\succeq 0$ and then set $U_i=u_i^2$, $V_i=v_i^2$, and $u_iv_i=0$ in $X_i$. It is clear that $X_i\succeq 0$ and $\mathrm{rank}(X_i)=1$ if and only if $u_iv_i=0$. Since $\mathrm{rank}(X_i)=1$ is nonconvex, we drop the rank-1 constraint to obtain the SDP relaxation: $x_i=u_i$, $u_i\geq 0$, $v_i\geq 0$, $X_i\succeq 0$.
>
> [4] Chiu et al. "Tight certification of adversarially trained neural networks via nonconvex low-rank semidefinite relaxations." ICML 2023.

---

> > ### Comment · Reviewer_zfFU · 2025-04-07
> >
> > Thanks for your detailed rebuttal and most of concerns are solved. I would like to keep my score unchanged.

---

### Official Review · Reviewer_4uhX · 2025-03-11

**Overall Recommendation:** 3

**Summary:**

The work proposes SDP-CROWN, a modified bound propagation framework based on CROWN for verifying the robustness of neural networks to $\ell_2$ norm perturbations. SDP-CROWN introduces an additional parameter into the bound propagation framework which is used for tightening the bias term in the linear relaxations propagated through the network. The formulation for the bias term is derived from the tighter Semidefinite Programming relaxation and enables a more accurate representation of the $\ell_2$-norm interdependencies between neurons. When compared to a number of competing approaches, the authors show experimentally that their method results in tighter bounds while preserving scalability.

**Claims And Evidence:**

- The general claims are mostly supported by sufficient evidence. SDP-CROWN clearly outperforms the baselines which are considered by the authors. However, as mentioned in the methods section, some comparisons with other methods are missing which makes it more difficult to assess the contributions of this work. Besides this, a clarification of what the "BM" method is (which outperforms SDP-CROWN) would be helpful.
- Figure 1: This figure is supposed to show the advantages of SDP-CROWN, but I find it confusing for a number of reasons. Firstly, it is unclear which dataset the ConvBig network used for creating this figure is actually trained on. Secondly, the caption refers to Section 6.1 for details but 6.1 does not contain any results on ConvBig. Thirdly, the axes in the figure are not labeled, it is unclear what the whiskers represent (confidence intervals? Or is the upper number the PGD upper bound of robustness?). Lastly, the numbers mentioned in the caption are different from the ones shown in the figure. Overall, this figure probably needs to be revised and currently doesn't help understand the method.

**Essential References Not Discussed:**

There seem to be a number of references on verification of neural networks to $\ell_2$ norm robustness missing in the related work section. The section only mentions SDPs and standard bound propagation. However, there are a number of other works on such perturbations, see e.g. Table 1 in [3] for a number of other papers that support $\ell_2$ perturbations. There are also works such as [4] which construct networks more amenable to robustness verification and which usually focus on $\ell_2$ robustness as well. However, neither of these feature in the related work section. Approaches such as [5] which compute the Lipschitz constant of a network using SDPs might also be relevant. A comparison with the method proposed by [6] which also supports $\ell_2$ perturbation would be extremely important since that method seems to perform comparably well when compared to bound propagation methods.

[3] Meng, M. H., Bai, G., Teo, S. G., Hou, Z., Xiao, Y., Lin, Y., & Dong, J. S. (2022). Adversarial robustness of deep neural networks: A survey from a formal verification perspective. IEEE Transactions on Dependable and Secure Computing.

[4] Hu, K., Zou, A., Wang, Z., Leino, K., & Fredrikson, M. (2023). Unlocking deterministic robustness certification on imagenet. Advances in Neural Information Processing Systems, 36, 42993-43011.

[5] Fazlyab, M., Robey, A., Hassani, H., Morari, M., & Pappas, G. (2019). Efficient and accurate estimation of lipschitz constants for deep neural networks. Advances in neural information processing systems, 32.

[6] Chiu, H. M., & Zhang, R. Y. (2023, July). Tight certification of adversarially trained neural networks via nonconvex low-rank semidefinite relaxations. In International Conference on Machine Learning (pp. 5631-5660). PMLR.

**Experimental Designs Or Analyses:**

In Table 1 there is a method termed "BM" for the MNIST-MLP which appears to outperform SDP-CROWN by a significant margin (although the runtime is much longer). Could the authors clarify what this method is and why it is not explained in the paper? I was surprised to see a method here which outperforms the proposed method but is not described in the work.

**Methods And Evaluation Criteria:**

- I wondered why the method is only integrated into the $\alpha$-CROWN verifier and not into $\alpha, \beta$-CROWN or even GCP-CROWN [1], possibly with the BICCOS cuts [2]. If this is difficult to do, at least a comparison with these should be possible. Since the authors describe at the beginning that this is not extremely difficult to do (just construct an $\ell_\infty$ box containing the $\ell_2$ box as a naive baseline), there seem to be little reason not to provide those results on the state-of-the-art verifiers (unless I'm missing something here).
- The fact that no benchmarking against SDP solvers is done despite the fact that the authors repeatedly mention that they are the most suitable method for solving $\ell_2$ robustness verification problems is a major weakness of the work. In Section 6, they state that "We ignore the comparison with SDP-based verifiers as we only use SDP relaxation for facilitating linear bound propagation over a $\ell_2$-norm ball region.". I don't understand this argument, to me the fact that SDP is only used in a certain way in the proposed approach doesn't mean that the proposed approach shouldn't be benchmarked against what the authors themselves identify as the (presently) most suitable way to solve the verification problem.

[1] Zhang, H., Wang, S., Xu, K., Li, L., Li, B., Jana, S., ... & Kolter, J. Z. (2022). General cutting planes for bound-propagation-based neural network verification. Advances in neural information processing systems, 35, 1656-1670.

[2] Zhou, D., Brix, C., Hanasusanto, G. A., & Zhang, H. (2024). Scalable Neural Network Verification with Branch-and-bound Inferred Cutting Planes. arXiv preprint arXiv:2501.00200.

**Other Comments Or Suggestions:**

#### Typos
- Line 113: tightened by optimizing over the linear relaxation themselves --> tightened by optimizing over the linear relaxation**s** themselves
- Line 160: defining a set of linear relaxation --> defining a set of linear relaxation**s**
- Line 170: why bound propagation tends to be lose --> why bound propagation tends to be lo**o**se
- Line 177: "with radii $\|x - \hat{x}\|$" - what norm is this? I assume $\ell_2$?
- Line 177: are a factor of $\sqrt{n}$ than the radius --> are a factor of $\sqrt{n}$ **larger** than the radius
- Line 244: the bound [...] satisfy --> the bound [...] satisf**ies**
- Line 253: We are now ready to proof --> We are now ready to pro**ve**
- Line 255: Fix any [...] and optimized each [...] yields --> Fix**ing** any [...] and optimiz**ing** each [...] yields
- Line 257: We show**s** that --> We show that
- Line 260: As shown in Lemma 5.1, linear lower bound [...] yields --> As shown in Lemma 5.1, **the** linear lower bound [...] yields
- Line 266: The desire result follows --> The desire**d** result follows
- Line 271: our method is guarantee to --> our method is guarantee**d** to
- Line 321: and $h(\alpha)$ form $\alpha$-CROWN --> and $h(\alpha)$ f**ro**m $\alpha$-CROWN
- Line 410: As neural network verification is **an** NP-hard, all methods --> As neural network verification is NP-hard, all methods
- Line 429: significantly narrowing the gap the PGD upper bound --> significantly narrowing the gap **with respect to** the PGD upper
bound

#### Other
- Can Figure 4 be moved to the main paper? It seems like there would be enough space for it and it would make it easier to check the figure while reading Section 6.3.

**Other Strengths And Weaknesses:**

The strengths and weaknesses are mentioned above.

**Questions For Authors:**

- Could the authors explain their reasoning in line 205ff a bit more? They assume that $\| \hat{x} \|_\infty \leq \rho$ holds and if it doesn't, they substitute something with $u_i^*, v_i^*$. Could they explain where exactly this $u_i^*, v_i^*$ are substituted and how that leads to their assumption being correct?
- Could the authors comment on GCP-CROWN/BICCOS and why they weren't considered in this work? Are any results on these verifiers available?
- Is there a reason why SDP-based verifiers are not evaluated?
- Are the authors aware of the substantial body of literature on $\ell_2$ robustness verification using special architectures such as [4] above? Could the compare their work/contextualise it relative to those works?

**Relation To Broader Scientific Literature:**

$\ell_2$ robustness is not considered in most of the verification literature or only considered in a somewhat naive way (except for the global robustness literature and the works on constructing networks with small Lipschitz constants), so this work tackles a problem that seems relevant. I find the lack of of contextualisation and comparison with these methods problematic.

**Theoretical Claims:**

I checked all the proofs in the main part of the paper. Some of them are a bit difficult to follow since a number of reformulation/rewriting steps are done implicitly, but overall the proofs seem correct to me.

---

> ### Author Rebuttal · Authors · 2025-03-31
>
> We thank you for the detailed review. We added **additional experiments on GCP-CROWN and BICCOS, as well as LipSDP [5] (another SDP method) for all models** as requested. We hope you can reevaluate our paper based on our response.
>
> **Clarifying Figure 1**
>
> We apologize for the confusion and will correct Figure 1. The model name should be ConvLarge, not ConvBig. ConvLarge is trained on CIFAR-10 and has 4 convolutional and 3 linear layers, totaling 24.6M parameters. The whiskers in Figure 1 illustrate the uncertainty gap where the true certified accuracy lies; blue numbers represent lower bounds on certified accuracy from different methods, and red numbers represent the PGD upper bound. A smaller gap indicates stronger verification.
>
> **Comparison to GCP-CROWN and BICCOS**
>
> During the rebuttal period, we implement $\ell_2$ norm support into GCP-CROWN and BICCOS. Particularly, the cutting planes found were using precise $\ell_2$ norm constraints rather than directly constructing an enclosing $\ell_\infty$ ball. This shows the best GCP-CROWN and BICCOS can do. We included comparisons to GCP-CROWN and BICCOS on **all benchmarks** in Table A below.
>
> [Table A: Comparisons to new baselines](https://imgur.com/a/QITgjmq)
>
> Since GCP-CROWN and BICCOS are all part of the $\alpha,\beta$-CROWN verifier, to avoid confusion, we renamed the original $\alpha,\beta$-CROWN baseline to $\beta$-CROWN (Wang et al., 2021). GCP-CROWN and BICCOS marginally improve over $\beta$-CROWN, although the gap to our method is still large especially on bigger models.
>
> **Explain BM method**
>
> BM is an SDP-based method [6]. It exactly solves the SDP relaxation of the verification problem using specialized low-rank solvers. BM is one of the tightest SDP-based verifiers (as demonstrated in their paper) but is not scalable to CIFAR-10 models. We choose BM because it is a recently published SDP-based method with strong results.
> BM is tighter than our method because our approach remains LP-based as the SDP relaxation is only used to refine the offset of linear bounds. However, our method is orders of magnitude more scalable; our ConvLarge is a factor of 100x larger than the models considered in BM. Our comparisons with BM demonstrate that the substantial improvement in scalability achieved by our method comes with only a mild loss of tightness.
>
> **Comparison to SDP-based methods**
>
> Besides the BM SDP-based method already reported in our paper, we conduct additional experiments on LipSDP [5] in Table A.
>
> [Table A: Comparisons to new baselines](https://imgur.com/a/QITgjmq)
>
> For a fair comparison, we run LipSDP with multiple configurations by splitting original networks into subnetworks. We show the results in Table A. The verified accuracy by LipSDP is significantly lower than ours, even in the tightest (slowest) setting with no split.
>
> We also considered additional SDP baselines that can scale to the networks we evaluated, such as SDP-FO (Dathathri et al., 2020). However, we found that their algorithm and implementation support $\ell_\infty$ norm only, and it is not straightforward to adapt their implementation to our setting.
>
> **Add new references**
>
> Thank you for bringing these references to our attention. We will incorporate these works into the final version to provide a more comprehensive discussion of related research. We’ve added an experimental comparison to GCP-CROWN [1], BICCOS [2], Lip-SDP [5] above, and [6] is the BM in our paper. We will cite and discuss [3] (a survey paper) and [4] (focusing on training certifiable networks with special architectures).
>
>
> **Response to Questions For Authors:**
>
> 1.  Given $c^T\textrm{ReLU}(x)\geq g^Tx+h$ that holds within $\Vert x-\hat x\Vert_\infty \leq \rho$. Observe that from bound propagation $g_i=c_i$ if $\hat x_i>\rho$, and $g_i=0$ if $\hat x_i<\rho$. Hence, $c_i\textrm{ReLU}(x_i)-g_ix_i=0$ if $|\hat x_i|>\rho$. Therefore $x_i^\star=\hat x_i$ is a minimizer of $\min_x c^T\textrm{ReLU}(x)-g^Tx\text{ s.t. } \Vert x-\hat x\Vert_2 \leq \rho$ if $|\hat x_i|>\rho$, and $x_i$ can be removed by substituting $x_i=\hat x_i$. It follows from positive/negative splitting $x_i^\star=u_i^\star - v_i^\star$ that $u_i^\star=\max\lbrace\hat x_i,0\rbrace$ and $v_i^\star=\min\lbrace\hat x_i,0\rbrace $.
>
> 2 & 3. See our comparison to GCP-CROWN, BICCOS, LipSDP and BM above.
>
> 4. Our method is a hybrid framework that tightens bound propagation using SDP relaxations specifically for ReLU activations. Our approach is complementary to existing work on leveraging specialized architectures, offering an alternative route to achieve certified robustness. We will add a paragraph to discuss methods that leverage specialized architectures, such as [4], Cayley layers, AOL, SLL, etc. While a direct comparison is not feasible due to the different aims of these approaches, we will explore the design of verification-friendly architectures that can utilize the strengths of both bound propagation and specialized architectures in future work.

---

> > ### Comment · Reviewer_4uhX · 2025-04-03
> >
> > I'd like to thank the authors for the detailed rebuttal that was provided, the clarifications are certainly useful. I am still slightly confused about the BM method and the fact that it is not mentioned in your paper but then appears as a benchmark in one table, but I am sure that this can be corrected. The new baselines comparing the work to GCP-CROWN and other SDP-based methods seem promising. Given this fact, I will raise my score.

---

> > > ### Author Response · Authors · 2025-04-07
> > >
> > > Thank you very much for raising the score. We will be sure to introduce all baselines (including SDP/BM) in the experiment section. We apologize for the confusion; we will add the below sentences to clarify the relationship between BM and SDP.
> > >
> > > > I am still slightly confused about the BM method and the fact that it is not mentioned in your paper but then appears as a benchmark in one table, but I am sure that this can be corrected.
> > >
> > > We wish to clarify that SDP is a *formulation*, whereas BM is a *solver*; it is one among several others such as MOSEK, SeDuMi, SDPNAL+, etc for solving SDP. To give an analogy, the system of equations $Ax=b$ is a formulation, whereas sparse Cholesky factorization, LU decomposition are all different solvers for computing the same $x$. Our intention was to compare against SDP, which as rightly stated in your review, is extremely important. In contrast, BM is only one of several possible SDP solvers (which we cited in the experimental section), and not really a "verification method" in its own right. Privately, we also tried solving using MOSEK for solving SDP, but found BM to be more scalable. We apologize that this distinction was not spelled out in the original submission nor in our rebuttal. Throughout the paper, all instances of "BM" should be understood to mean "SDP", with BM being only the solver used to solve it.
> > >
> > > We also added comparisons with LipSDP. This, like our proposed method, is an "SDP-based verification method", because it uses the ideas of SDP but does not actually solve it.
> > >
> > > Please let us know if there are further concerns that we should address. If you are happy with our rebuttals/answers, we would appreciate any further score adjustments. Thank you very much.

---

### Official Review · Reviewer_4cg2 · 2025-03-14

**Overall Recommendation:** 4

**Summary:**

The authors propose a new method they call SDP-CROWN. They use the framework of semidefinite programming to derive linear bounds on the networks behaviour however based not on an $L_\infty$ norm but based on $L_2$ norm bound perturbations. This linear bound can then be used for any linear bound propagation method to obtain a certificate. Specifically, what they do is to use a standard bound propagation method, but the constant offset of that bound is then adjusted (from an $L_\infty$ bound to and $L_2$ bound offset) but still sound.

**Claims And Evidence:**

- They claim that their bounds can be up to $\sqrt(n)$ better compared to standard single-neuron bounds. While this is clear for the input - for the output this can be larger or smaller i would assume. (imprecisions can accumulate)
- The authors claim also soundness which is supproted by a proof (4.1).

**Essential References Not Discussed:**

- https://openreview.net/pdf?id=awHTL3Hpto I have the feeling that the limits regarding the most optimal linear bounds (Salman et al 2019) regarding the expressivity also apply here. Specifically, just restricting $\ell_\infty$ to $\ell_2$ and adjustment of the offset do not address fundamental expressivity shortcomings.

**Experimental Designs Or Analyses:**

The experimental design is standard. That said, i would be curious how far one could push the method.

**Methods And Evaluation Criteria:**

The method is evaluated against several competitors that are tailored for $L_\infty$. Evaluation takes place on MNIST and CIFAR-10 which is standard. The creteria are certified accuracy and time.

**Other Comments Or Suggestions:**

No further suggestions.

**Other Strengths And Weaknesses:**

I like the combination of SDP and linear bound propagation. This is a neat idea - though it is not quite clear to me how far one could push this idea.

**Questions For Authors:**

The question are above.

**Relation To Broader Scientific Literature:**

Most related work i know of is adequadely discussed. What i find curious about the work here is that the offset adjustmeht here is the opposite of what is done in [A], where unsound bounds are adjusted to be sound. Here, sound bounds are adjusted to be more tight but still sound.

- [A] https://proceedings.neurips.cc/paper_files/paper/2019/file/f7fa6aca028e7ff4ef62d75ed025fe76-Paper.pdf

**Theoretical Claims:**

See above.

---

> ### Author Rebuttal · Authors · 2025-03-31
>
> We want to thank reviewer 4cg2 for valuable comments and for recognizing the key contributions of our paper. We hope our response would adequately address your questions and concerns.
>
>  **"They claim that their bounds can be up to $\sqrt{n}$ better compared to standard single-neuron bounds. While this is clear for the input - for the output this can be larger or smaller i would assume. (imprecisions can accumulate) "**
>
> The input bound improvement by a factor of $\sqrt{n}$ follows directly from the fact that the volume of an $\ell_2$ ball is smaller than that of an $\ell_\infty$ box by this factor.
>
> Regarding the output bound, we prove that our method can achieve up to a $\sqrt{n}$ improvement vs traditional bound propagation (Theorem 5.2). However, this proof applies specifically to as simple network $\mathrm{ReLU(x)}$ under $\ell_2$ perturbations of the form $\Vert x\Vert_2\leq\rho$.
>
> For general multi-layer neural networks, the extent of improvement is hard to quantify theoretically. To address this, we provide empirical results demonstrating that our method consistently outperforms bound propagation methods, especially on large neworks.
>
> **"The experimental design is standard. That said, I would be curious how far one could push the method.
> "**
>
> Thank you for your insightful comment. In this paper, we utilize SDP to tighten the offset of the linear bounds in bound propagation. Our method has been integrated into $\alpha$-CROWN with minimal overhead, as it only introduces one additional variable $\lambda$ per postactivation neuron.
>
> Our next step is to integrate our method into the $\alpha,\beta$-CROWN codebase, hence allowing our method to work with powerful branch-and-bound based methods and a wide range of neural network architecture. Our current algorithm already outperformed the existing branch-and-bound based approaches (including GCP-CROWN, BICCOS and beta-CROWN, as presented in the **[new table](https://imgur.com/a/QITgjmq)** we added during rebuttal), and additional improvements can be further demonstrated once we finish this integration. Additionally, we aim to extend our approach to jointly optimize both the slope and offset terms, with the goal of developing optimal linear relaxations for $\ell_2$ perturbations.  We expect these two future directions to yield tighter results on an even larger scale than considered in the present paper.
>
> **"Most related work I know of is adequately discussed. What I find curious about the work here is that the offset adjustment here is exactly the opposite of what is done in [A] and [B], where unsound bounds are adjusted to be sound. Here, sound bounds are adjusted to be more tight but still sound."**
>
> We appreciate the reviewer for bringing these references to our attention. We will include a discussion of these works in the final version of our paper.
>
> [A] presents a method for certifying robustness against geometric transformations. This approach begins with potentially unsound bounds and subsequently adjusts them to be sound through a combination of sampling and optimization techniques. In contrast, our method operates in the opposite manner; we start with sound bounds and refine them to be tighter while ensuring they remain valid.
> Regarding [B], we believe this reference may have been included by mistake, as it presents a framework for robotic tissue manipulation using deep learning for feature extraction, which does not appear relevant to the context of our work. We will be happy to discuss any additional works you suggest.
>
> **"I have the feeling that the limits regarding the most optimal linear bounds (Salman et al 2019) regarding the expressivity also apply here. Specifically, just restricting $\ell_2$ to $\ell_\infty$ and adjustment of the offset do not address fundamental expressivity shortcomings."**
>
> Our critical contribution is to introduce multi-neuron SDP relaxations into bound propagation. This allowed us to overcome the “convex relaxation barrier” faced by traditional single-neuron LP relaxations underlying the vast majority of prior work. It also improved upon prior work on multi-neuron relaxations, which were either too loose (i.e. LP-based) or limited to tiny models (i.e. SDP-based). Addressing $\ell_2$-norm constraints is a crucial first step, but we anticipate further work to greatly expand the applicability of models and new threat models (beyond $\ell_p$ norm) using our key idea of combining SDP with bound propagation. We will also cite the paper on the Expressivity of ReLU networks in our final version.

---

### Decision · Program_Chairs · 2025-05-01

**Decision:**

Accept (spotlight poster)

**Comment:**

This paper combines the two techniques: CROWN (linear bound propagation) and semi-definite programming based bound propagation. The claims are well supported both empirically and theoretically. The reviewers have provided details feedback on how to improve the related work section, which authors should take into account while preparing the final version.